# Navigating the Flatlands: Dual Adaptive Sharpness-Aware Minimization for Domain Generalization

Junwen He [1]  Yang He [1]  Lebing Zheng [1]  Zirui Yin [1]  Hong-Yu Zhang [1]  Yulong Wang [1]

## Abstract

Finding flat minima in the loss landscape is a key strategy for Domain Generalization (DG). However, its effectiveness is often limited by two crucial challenges. 1) Domain Shift: Existing methods like Sharpness-Aware Minimization (SAM) apply a uniform optimization strategy across all domains, overlooking the differences of the learning difficulties among multiple domains and thus performing poorly on challenging domains. 2) Anisotropic Sharpness: By perturbing parameters along a single gradient direction, SAM and its variants ignore multi-directional flatness, making the model converge to minima that remain sharp in other directions. The combined challenges make it more difficult for the model to find truly robust solutions in multi-domain scenarios. To overcome these limitations, we propose the Dual Adaptive Sharpness-Aware Minimization (DA-SAM), which comprises two key modules: Dynamic Adaptive Scaling (DAS) module and Adaptive Multi-Directional Flattening (AMDF) module. First, to tackle the domain shift problem, the DAS module computes the real-time loss on each domain to adaptively generate domain-specific scaling factors that guide the generation of perturbation directions. Second, the AMDF module calculates local flatness by generating multiple directions to simulate perturbations in the parameter space. Based on the learned local flatness metric, it dynamically adjusts the perturbation step size to guide the model parameters to be away from anisotropic sharp regions. Crucially, DAS provides domain-level guidance that makes AMDF's multi-directional geometric exploration more targeted and effective. Extensive

experiments on five DG benchmarks demonstrate the effectiveness of our DA-SAM algorithm.

## 1. Introduction

Deep neural networks have achieved remarkable success on i.i.d. (independent and identically distributed) data, where the training and testing data are drawn from the same underlying distribution (Krizhevsky et al., 2017; He et al., 2016). However, they often falter when deployed in the real world due to domain shifts, a phenomenon where the testing data distribution differs from the training data (Torralba & Efros, 2011; Wang et al., 2022). For instance, a model trained to recognize animals in clear photographs may fail when deployed on artistic paintings or sketches. Domain Generalization (DG) aims to tackle this fundamental challenge by learning a single model from multiple source domains that generalizes robustly to unseen target domains (Blanchard et al., 2011; Wang et al., 2022). A particularly promising line of research connects generalization to the geometry of the loss landscape, suggesting that models residing in flat minima—wide, low-loss valleys—are inherently more robust (Hochreiter & Schmidhuber, 1997; Keskar et al., 2017).

Methods based on Sharpness-Aware Minimization (SAM) (Foret et al., 2021) operationalize this idea by seeking such flat regions, and have shown significant potential in improving generalization (Cha et al., 2021). However, its effectiveness in multi-domain scenarios is often limited by two crucial challenges. 1) **Domain Shift**: Existing sharpness-aware methods apply a uniform optimization strategy across all domains. This one-size-fits-all approach fails to consider the significant differences in learning difficulty among them. As a result, the model tends to underfit the more challenging domains while over-specializing on the easier ones. This is a critical issue in standard DG datasets like PACS, where the photo domain is demonstrably less complex than sketch domain (Li et al., 2017). 2) **Anisotropic Sharpness**: SAM and its variants perturb parameters along a single gradient direction. This approach inherently ignores the anisotropy of the loss landscape, causing the model to converge to minima that may be flat along this one direction but remain sharp in others. Such a one-directional view of flatness is deceptive.

[1]Hubei Key Laboratory of Agricultural Bioinformatics, College of Informatics, Huazhong Agricultural University, Hongshan District, Wuhan City, Hubei Province, China. Correspondence to: Yulong Wang <wangyulong6251@gmail.com>.

A minimum that appears to be a flat valley from one perspective can actually be a sharp canyon from other directions, and this severely undermines the robustness of the model to domain shifts (Chaudhari et al., 2019).

Consequently, the combined limitations make it more difficult for the model to find robust solutions. A uniform search in a single direction is poorly equipped to navigate a landscape with varying difficulty levels and anisotropic geometry. To overcome these limitations, we propose a synergistic optimization algorithm, called **Dual Adaptive Sharpness-Aware Minimization (DA-SAM)**, which comprises a Dynamic Adaptive Scaling (DAS) module and an Adaptive Multi-Directional Flattening (AMDF) module.

First, to tackle domain shift, the **DAS** module computes the real-time loss in each domain to adaptively generate domain-specific scaling factors. These factors guide the generation of domain-aware perturbation directions, ensuring that harder domains are explored more cautiously. Second, the **AMDF** module is designed to address anisotropic sharpness. It calculates local flatness by generating multiple directions to probe the geometry of the parameter space. Based on the local flatness metric, the module dynamically adjusts the perturbation step size. This final step actively guides the model parameters away from anisotropic sharp regions and towards more robust, isotropically flat minima. Crucially, DAS provides domain-level guidance that makes AMDF's multi-directional geometric exploration more targeted and effective. Extensive experiments on five DG benchmarks demonstrate the effectiveness of our DA-SAM algorithm.

## 2. Related work

### 2.1. Domain Generalization

The field of Domain Generalization (DG) has explored several primary strategies to tackle distribution shifts. Early approaches focused on Domain Alignment (Ben-David et al., 2010), aiming to learn domain-invariant features by minimizing statistical discrepancies between source domains (Li et al., 2018a; Sun & Saenko, 2016). Other popular paradigms include Meta-Learning (Finn et al., 2017), which simulates domain shifts during training (Li et al., 2018a), and Data Augmentation (Volpi et al., 2018; Zhou et al., 2021), which enriches training data with synthesized samples.

A distinct line of research directly leverages gradient information. These methods operate on the premise that conflicting gradients across domains are the primary cause of overfitting to domain-specific features, and thus seek parameter regions where gradients from different domains naturally align (Arjovsky et al., 2019).

While these methods address crucial aspects of DG, an-other fundamental approach seeks to make the model robust through the geometry of the loss landscape. This perspective, centered on finding flat minima, is the focus of our work and is complementary to the aforementioned strategies.

### 2.2. Sharpness-Aware Optimization

The principle that flatter minima yield better generalization (Hochreiter & Schmidhuber, 1997; Keskar et al., 2017) was effectively translated into a concrete optimization strategy by Sharpness-Aware Minimization (SAM) (Foret et al., 2021). SAM seeks flat regions by minimizing the maximum loss within a local neighborhood, found via a one-step gradient ascent approximation. This has spurred a rich body of follow-up work, which can be broadly categorized along several research thrusts.

One line of research aims to find flatness by proposing better measures. For instance, Surrogate Gap Guided Sharpness-Aware Minimization (GSAM) (Zhuang et al., 2022) introduced a surrogate gap, while other methods used the gradient norm (Zhang et al., 2023b) or explicitly regularized the Hessian trace (Wu et al., 2024). To account for parameter scale and manifold geometry, Adaptive Sharpness-Aware Minimization (ASAM) (Kwon et al., 2021) adjusts the perturbation radius proportionally to weight magnitudes, while Fisher Sharpness-Aware Minimization (Fisher SAM) (Kim et al., 2022) utilizes the Fisher information matrix to align the neighborhood with the statistical manifold. More recently, Flatness-Aware Minimization for Domain Generalization (FAD) (Zhang et al., 2023a) proposed a unified framework to simultaneously optimize both zeroth-order (loss value) and first-order (gradient norm) flatness. Another direction focuses on the update mechanism itself. For example, Friendly Sharpness-Aware Minimization (FSAM) (Li et al., 2024) decomposed the gradient to isolate and leverage beneficial stochastic noise, while Sharpness-Aware Gradient Matching (SAGM) (Wang et al., 2023) encouraged convergence to flatter minima by explicitly aligning the gradients of the original and the perturbed loss functions. Furthermore, SAM has been adapted to handle specific data distributions; notably, Class-Conditional Sharpness-Aware Minimization (CC-SAM) (Zhou et al., 2023) and Focal Sharpness-Aware Minimization (Focal-SAM) (Li et al., 2025) implement class-conditional and focal sharpness penalties to mitigate performance degradation in long-tailed recognition. Despite these significant advances, a critical gap remains in their application to DG. All these methods—whether they refine the flatness metric (like FAD, ASAM, and Fisher SAM) or the update mechanism (like FSAM, SAGM, and Focal-SAM)—fundamentally apply a uniform optimization strategy. They treat all source domains as a monolith, failing to adapt to the varying learning difficulties caused by domain shift. In addition, they ultimately rely on a single gradient direction to define their search path. This approach does not explic-

itly address the problem that this path might be part of an anisotropic landscape, where flatness in one direction does not guarantee robustness in all others.

To our knowledge, DA-SAM is the first algorithm designed to address both of the fundamental limitations simultaneously. It pairs a synergistic mechanism to handle the domain shift and anisotropic sharpness for seeking a more comprehensively robust solution for domain generalization.

## 3. Preliminaries

### 3.1. Problem Formulation

In the multi-source Domain Generalization (DG) setting, we are given $K$ source domains $\mathcal{D} = \{\mathcal{D}_k\}_{k=1}^K$. Let $\theta \in \mathbb{R}^d$ denote the parameters of a neural network $f(\cdot; \theta)$. The empirical risk is defined as the average loss over all source domains:

$$L(\theta; \mathcal{D}) = \frac{1}{K} \sum_{k=1}^K L_k(\theta), \tag{1}$$

$$\text{where} \quad L_k(\theta) = \mathbb{E}_{(x,y) \sim \mathcal{D}_k}[\ell(f(x; \theta), y)].$$

Here $\ell(\cdot, \cdot)$ is a task-specific loss function, such as cross-entropy. The goal of DG is to learn parameters $\theta$ that not only minimize the training risk $L(\theta; \mathcal{D})$, but also generalize well to an unseen target domain $\mathcal{D}_{\text{target}}$. For notational simplicity, we will omit the dependency on $\mathcal{D}$ and write $L(\theta)$ instead of $L(\theta; \mathcal{D})$ in the subsequent sections, unless otherwise specified.

### 3.2. Sharpness-Aware Minimization (SAM)

To improve generalization, Sharpness-Aware Minimization (SAM) (Foret et al., 2021) reframes the optimization goal. Instead of finding a single point $\theta$ with the lowest loss, it aims to find a parameter region that is uniformly flat, i.e., the loss remains low even when the parameters are slightly perturbed. This is formalized by a min-max objective:

$$\min_\theta \max_{\|\epsilon\|_2 \leq \rho} L(\theta + \epsilon), \tag{2}$$

where $\epsilon$ represents the adversarial perturbation vector, and the radius $\rho$ defines the size of the neighborhood in which the worst-case perturbation is sought. Since solving this inner maximization exactly is intractable, SAM approximates the perturbation $\hat{\epsilon}(\theta)$ using a single step of gradient ascent, which is both efficient and effective:

$$\hat{\epsilon}(\theta) \approx \rho \frac{\nabla L(\theta)}{\|\nabla L(\theta)\|_2}. \tag{3}$$

The optimizer then minimizes the loss at this perturbed point, $L(\theta + \hat{\epsilon}(\theta))$, effectively steering the parameters $\theta$ towards flatter regions. It is crucial to note a key characteristic

of this procedure for the DG context: the perturbation $\hat{\epsilon}(\theta)$ is computed using the single, aggregated gradient $\nabla L(\theta)$, which is an average over all source domains. This results in a perturbation that is inherently **uni-directional** (pointing in only one direction) and **domain-agnostic** (treating the contribution of all domains as a monolith). As we will argue in the next section, these two properties—its reliance on a single direction and its monolithic treatment of all domains—are the direct causes of the **Anisotropic Sharpness** and **Domain Shift** challenges, respectively.

## 4. The Proposed Method: DA-SAM

The proposed Dual Adaptive Sharpness-Aware Minimization (DA-SAM) enhances standard sharpness-aware optimization to address the challenges of domain shift and anisotropic sharpness outlined in Section 3.2. The overall architecture is illustrated in Figure 1. It comprises two synergistic modules: a Dynamic Adaptive Scaling (DAS) module that provides domain-level guidance, and an Adaptive Multi-Directional Flattening (AMDF) module that performs a multi-directional geometric exploration.

### 4.1. Comparison with Prior Works

While SAM (Foret et al., 2021) and its variants have improved generalization, their application to DG is hindered by two fundamental limitations discussed previously: **Domain Shift** (leading to a suboptimal uniform strategy) and **Anisotropic Sharpness** (due to the reliance on a single perturbation direction). To overcome these limitations, DA-SAM is proposed by a dual-adaptive algorithm. Table 1 conceptually positions our algorithm against prior works by features. To highlight our unique approach, Table 2 contrasts DA-SAM's core mechanism with existing methods.

### 4.2. Dynamic Adaptive Scaling (DAS)

**Motivation.** A uniform optimization strategy is suboptimal for DG due to varying learning difficulties across domains. DAS addresses this by adaptively allocating an exploratory budget based on each domain's real-time loss, enabling a more robust search.

**Formulation and Analysis.** DAS computes a dynamic, scale-invariant scaling factor $\sigma_k$ for each domain $\mathcal{D}_k$ using a reversed sigmoid function on the standardized loss:

$$\sigma_k = s_{\min} + (s_{\max} - s_{\min}) \left( 1 - \sigma_{\text{sig}} \left( C \cdot \frac{L_k - \bar{L}}{\text{std}(L) + \epsilon} \right) \right), \tag{4}$$

where $\sigma_{\text{sig}}$ is the sigmoid function. $\bar{L}$ and $\text{std}(L)$ are the mean and standard deviation of the set of per-domain losses $\{L_1, \ldots, L_K\}$ computed on the current mini-batch. The terms $s_{\min}$ and $s_{\max}$ serve as **boundary constants**, while $C$ acts as a **scaling coefficient**. To ensure practical usability,

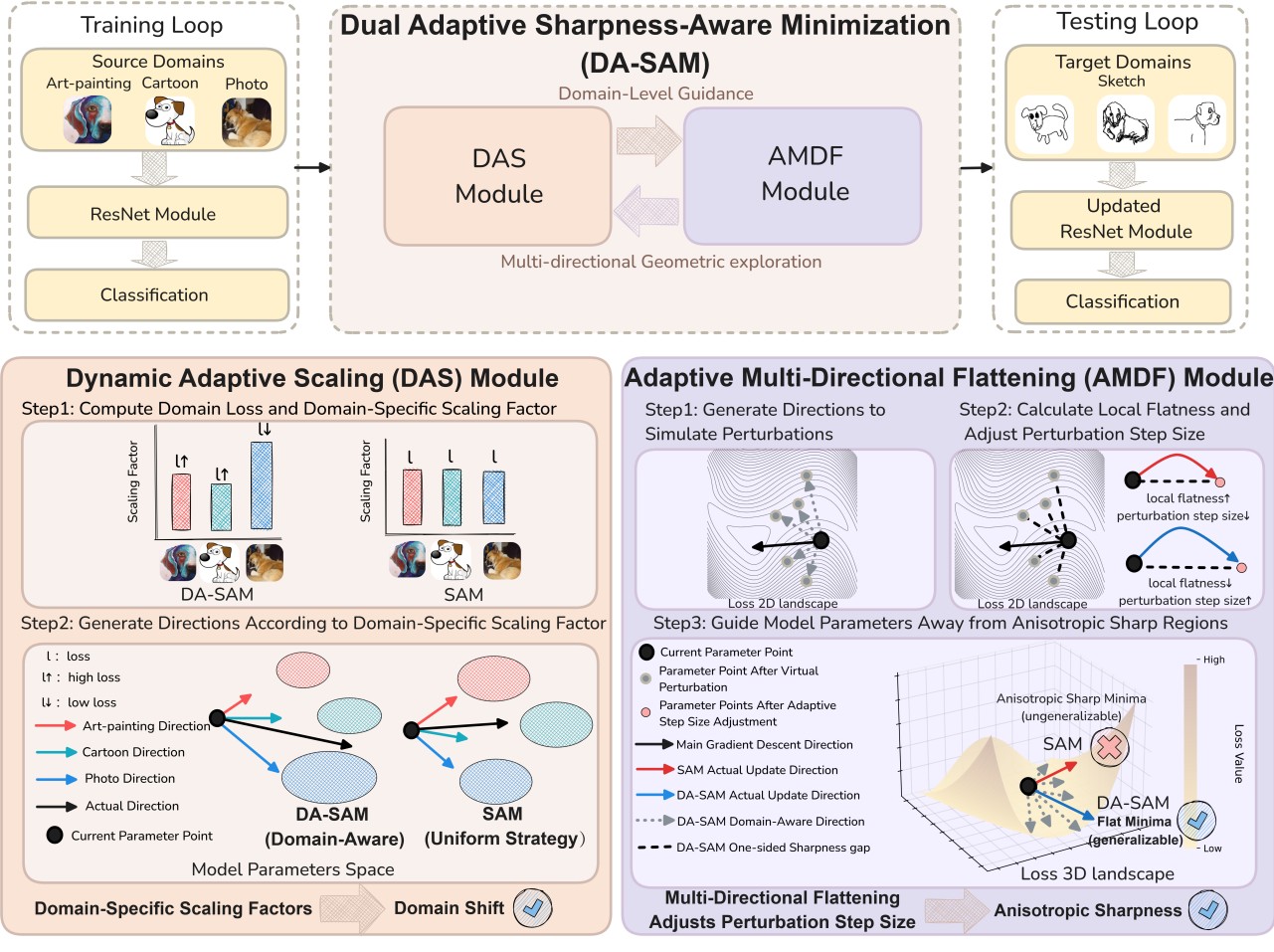

*Figure 1.* The overall architecture of the DA-SAM. The framework operates within a standard **training and testing loop** (top row). The core of our method is the **Dual Adaptive Sharpness-Aware Minimization** procedure (middle), which consists of two synergistic modules detailed below. The **Dynamic Adaptive Scaling (DAS) module** (bottom left) computes domain-specific scaling factors based on real-time losses to address domain shift. Guided by these factors, the **Adaptive Multi-Directional Flattening (AMDF) module** (bottom right) calculates a local flatness metric, and adaptively adjusts the perturbation step size to handle anisotropic sharpness. The final optimized model is then evaluated on an unseen target domain.

we fix them to robust default values across all benchmarks. To maintain a consistent total perturbation budget across domains, we further **normalize the scaling factors** $\tilde{\sigma}_k$ obtained from Eq. (4) as: $\sigma_k = (\tilde{\sigma}_k / \sum_{j=1}^{K} \tilde{\sigma}_j) \times K$. $\epsilon$ is a small constant for numerical stability. Eq. (4) is based on the intuition that exploration should adapt to learning difficulty. A domain with a large loss $L_k$ receives a small $\sigma_k$ to encourage a focused search, while a low loss domain is assigned a large $\sigma_k$ to promote broader exploration. Due to this relative normalization, individual scaling factors can exceed 1.0 (as shown in Fig. 2), effectively redistributing the exploration budget from well-learned domains to more challenging ones while keeping the average scaling at 1.0 to align with standard SAM.

**Theoretical Justification.** The DAS mechanism can be

viewed as an implicit solver for the robust Distributionally Robust Optimization (DRO) objective (Sagawa et al., 2020):

$$\min_{\theta} \max_{k} \mathcal{L}_k(\theta). \qquad (5)$$

Consider a gradient-based update at step $t$. To minimize the objective in Eq. (5), the optimization must effectively reduce the loss of the current worst-case domain, $k^* = \arg\max_k \mathcal{L}_k(\theta_t)$. The change in this loss after one step is approximately:

$$\mathcal{L}_{k^*}(\theta_{t+1}) \approx \mathcal{L}_{k^*}(\theta_t) - \eta_t \langle \nabla \mathcal{L}_{k^*}(\theta_t), g_t \rangle, \qquad (6)$$

where $\eta_t$ is the learning rate and $g_t$ is the update direction used by the optimizer. The core challenge is that computing the per-domain gradient $\nabla \mathcal{L}_k(\theta_t)$ for each domain at every

*Table 1.* Comparison of different related works.

| Algorithm | Task | Multi-direction | Domain-Aware | Dual-Adaptive |
|---|---|---|---|---|
| SAM (Foret et al., 2021) | Generalization | ✗ | ✗ | ✗ |
| GSAM (Zhuang et al., 2022) | Generalization | ✗ | ✗ | ✗ |
| SAGM (Wang et al., 2023) | Domain Generalization | ✗ | ✗ | ✗ |
| GAM (Zhang et al., 2023b) | Generalization | ✗ | ✗ | ✗ |
| FAD (Zhang et al., 2023a) | Domain Generalization | ✗ | ✗ | ✗ |
| CRSAM (Wu et al., 2024) | Generalization | ✗ | ✗ | ✗ |
| FSAM (Li et al., 2024) | Generalization | ✗ | ✗ | ✗ |
| **DA-SAM (ours)** | **Domain Generalization** | ✓ | ✓ | ✓ |

*Table 2.* Comparison of core mechanisms in SAM and its variants. Notation: $w$: parameters; $L(w)$: loss; $\nabla L(w)$: gradient; $L_p(w)$: perturbed loss; $L_B(w)$: bounded loss; $h_i$: one-sided sharpness gap; $H(w)$: Hessian; $\mathcal{B}$: neighborhood ball; $\text{Tr}(\cdot)$: trace of a matrix; $R^{(0/1)}$: zeroth/first-order flatness; $\mathbb{E}[\cdot]$, $\text{Var}[\cdot]$: expectation and variance.

| Algorithm | Mathematical Object | Primary Focus | Key Differentiator |
|---|---|---|---|
| SAM (Foret et al., 2021) | $\nabla L(w)$ | Sharpness Metric | Gradient Ascent |
| GSAM (Zhuang et al., 2022) | $L_p(w) - L(w)$ | Loss Objective | Surrogate Gap |
| SAGM (Wang et al., 2023) | $(\nabla L_p(w), \nabla L(w))$ | Gradient Alignment | Perturbation Consistency |
| GAM (Zhang et al., 2023b) | $\|\nabla L_{\mathcal{B}}(w)\|_2$ | Sharpness Metric | Gradient Norm Minimization |
| FAD (Zhang et al., 2023a) | $R^{(0)}(w) + (1-\alpha)R^{(1)}(w)$ | Loss Objective | Zeroth + First-Order Flatness |
| CRSAM (Wu et al., 2024) | $\text{Tr}(H(w))$ | Loss Objective | Curvature Regularization |
| FSAM (Li et al., 2024) | $\nabla L_B(w) - \nabla L(w)$ | Perturbation Direction | Friendly Perturbation |
| **DA-SAM (ours)** | $\mathbb{E}[h_i] + \sqrt{\text{Var}[h_i]}$ | **Procedural Adaptation** | **Magnitude Control** |

step would be computationally prohibitive. Therefore, following the standard practice in SAM-based methods, our approach relies on the aggregated gradient computed from a single perturbed point. The perturbation directions in the subsequent AMDF module (Section 4.3) for domain $k$ are generated with a domain-specific variance controlled by $\sigma_k$. The scaling factor $\sigma_k$ from DAS directly controls the noise variance during geometric exploration. For the worst-case domain $k^*$, Eq. (4) prescribes a small $\sigma_{k^*}$. It leads to a tighter sampling of perturbation directions around the main gradient, resulting in a more stable and reliable estimate of the update direction $g_t$ with respect to the worst-case domain. This focused exploration ensures that the update step (Eq. (6)) makes meaningful progress on reducing the maximum risk. Thus, DAS acts as a gradient-based mechanism for worst-case optimization. The implicit regularization provides the crucial domain-level guidance for AMDF module.

### 4.3. Adaptive Multi-Directional Flattening (AMDF)

**Motivation.** The uni-directional perturbation of standard SAM provides an incomplete picture of the loss landscape's geometry. A truly robust minimum should be isotropically flat, exhibiting low sharpness uniformly across all directions. To quantify this, we require a metric that captures not only the average sharpness but also its directional consistency.

AMDF is designed to compute such a metric and use it to procedurally guide the optimizer.

**Formulation and Analysis.** The AMDF procedure consists of three main steps. First, it probes the landscape using a set of $N$ diverse directions for each domain, leveraging the guidance from the DAS module. Let $\hat{\mathbf{g}}(\theta) = \nabla L(\theta)/\|\nabla L(\theta)\|_2$ be the normalized aggregated gradient. We construct a distribution of gradient-biased random directions $\{\mathbf{d}_{i,k}(\theta)\}_{i=1}^{N}$ for each domain:

$$\mathbf{d}_{i,k}(\theta) = \frac{\hat{\mathbf{g}}(\theta) + \sigma_k \cdot \mathbf{z}_i}{\|\hat{\mathbf{g}}(\theta) + \sigma_k \cdot \mathbf{z}_i\|_2}, \quad \text{where } \mathbf{z}_i \sim \mathcal{N}(0, \mathbf{I}). \quad (7)$$

Here $\sigma_k$ is the domain-specific diversity factor provided by the DAS module (Eq. (4)). The factor is the mathematical embodiment of the synergy between our two modules. Second, for each generated direction, we compute the **one-sided sharpness gap**:

$$h_{i,k}(\theta) = \max(0, L(\theta + \rho \cdot \mathbf{d}_{i,k}(\theta)) - L(\theta)), \quad (8)$$

where $\rho$ is the perturbation radius. Third, inspired by risk-averse optimization (Shapiro et al., 2009), we define **Anisotropic Sharpness Metric**, $L_{\text{AMDF}}(\theta)$:

$$L_{\text{AMDF}}(\theta) = \mathbb{E}_{i,k}[h_{i,k}(\theta)] + \sqrt{\text{Var}_{i,k}[h_{i,k}(\theta)]}. \quad (9)$$

This metric captures both the expected sharpness ($\mathbb{E}[h]$) and its directional volatility ($\sqrt{\mathrm{Var}[h]}$). The use of the standard deviation ensures that both terms are dimensionally consistent, allowing for a stable and principled aggregation of the geometric risk.

**Theoretical Connection to Hessian Spectrum.** The validity of $L_{\mathrm{AMDF}}$ as a metric for isotropic flatness can be justified by its connection to the Hessian matrix $H(\theta)$. Near a local minimum $\theta^*$, a second-order Taylor expansion of the loss gives:

$$L(\theta^* + \epsilon) \approx L(\theta^*) + \nabla L(\theta^*)^T \epsilon + \frac{1}{2} \epsilon^T H(\theta^*) \epsilon. \quad (10)$$

Since $\nabla L(\theta^*) = 0$ at a minimum and letting $\epsilon = \rho \cdot \mathbf{d}$, the sharpness gap $h(\theta^*)$ for a direction $\mathbf{d}$ is approximately:

$$h(\theta^*) \approx \frac{1}{2} \rho^2 \mathbf{d}^T H(\theta^*) \mathbf{d}. \quad (11)$$

For analyzing the effect of the random component, we consider the properties of noise $\mathbf{z}_i \sim \mathcal{N}(0, \mathbf{I})$, for which $\mathbb{E}[\mathbf{z}_i] = 0$ and $\mathbb{E}[\mathbf{z}_i \mathbf{z}_i^T] = \mathbf{I}$. Following the principles of stochastic trace estimation (Yao et al., 2020), the expected sharpness gap is proportional to the trace of the Hessian:

$$\mathbb{E}[h(\theta^*)] \approx \frac{1}{2} \rho^2 \mathbb{E}[\mathbf{d}^T H \mathbf{d}] = \frac{1}{2} \rho^2 \mathrm{Tr}(H \mathbb{E}[\mathbf{d}\mathbf{d}^T])$$
$$= \frac{1}{2} \rho^2 \mathrm{Tr}(H). \quad (12)$$

Here $\mathrm{Tr}(H) = \sum_j \lambda_j$, is the sum of its eigenvalues, representing the average curvature. Thus, minimizing $\mathbb{E}[h]$ corresponds to minimizing the average sharpness. Furthermore, the variance of the sharpness gap relates to the variance of the Hessian's eigenvalues. A perfectly isotropic minimum would have all non-zero eigenvalues equal ($\lambda_1 = \lambda_2 = \cdots = \lambda_n$), resulting in zero variance. High variance in the sharpness gap $h$ implies a large variance in the directional curvatures, which is a hallmark of anisotropy. Therefore, by penalizing both the mean and the standard deviation of the sharpness gap in our $L_{\mathrm{AMDF}}$ metric, we are implicitly promoting minima where the Hessian eigenvalues are not only small on average, but also have a tight distribution. This directly encourages the model to converge to an **isotropically flat** region.

**Final Procedural Adaptation.** Finally, AMDF uses the metric to procedurally modulate the perturbation step size. We start with a dynamic base step size, $Step_{\mathrm{base}} = \rho/(\|\nabla L(\theta)\|_2 + \epsilon)$, which is inversely proportional to the gradient norm (where $\rho$ is a radius hyperparameter and $\epsilon$ is a small constant for numerical stability). The final adaptive step size $Step_{\mathrm{DA\text{-}SAM}}$ is then determined by a hyperparameter-free rational function:

$$Step_{\mathrm{DA\text{-}SAM}} = \frac{Step_{\mathrm{base}}}{1 + L_{\mathrm{AMDF}}(\theta)}. \quad (13)$$

The formulation ensures that the step size smoothly decreases as the geometric risk $L_{\mathrm{AMDF}}$ increases, thus promoting a cautious optimization in sharp or anisotropic regions.

### 4.4. The Full DA-SAM Algorithm and Objective

The complete DA-SAM procedure integrates the DAS and AMDF modules into a cohesive, multi-stage update at each training step, as detailed in Algorithm 1. The process begins by sensing domain difficulty to generate adaptive scaling factors $\{\sigma_k\}$ via the DAS module. Guided by these factors, the AMDF module then conducts a domain-aware geometric probing to compute the anisotropic sharpness metric, $L_{\mathrm{AMDF}}$. This metric is subsequently used to modulate the final perturbation step size, $Step_{\mathrm{DA\text{-}SAM}}$. Finally, the final perturbation vector, $\epsilon_w$, is constructed by scaling the uniformly averaged direction of all probe directions, i.e., $\epsilon_w = Step_{\mathrm{DA\text{-}SAM}} \cdot (\frac{1}{NK} \sum_{i,k} \mathbf{d}_{i,k})$, before updating the model parameters. This dual-adaptive procedure transforms the standard sharpness-aware update into an intelligent, geometry-aware navigation of the loss landscape, allowing DA-SAM to seek minima that are both isotropically flat and robust to domain shifts.

---

**Algorithm 1 :** The DA-SAM Algorithm

---

**Initialize:** Model parameters $\theta_0$.
**Input:** Training domains $\mathcal{D} = \{\mathcal{D}_k\}_{k=1}^K$, loss function $\ell$, base step size $\rho_{\mathrm{base}}$, number of directions $N$, total iterations $T$.

1: **for** $t = 0$ **to** $T - 1$ **do**
2:     Sample a mini-batch of data with domain labels;
3:     Compute per-domain loss $L_k(\theta_t)$;
4:     Compute scaling factor $\sigma_k$ according to Eq. (4);
5:     Generate $N$ probe directions $\mathbf{d}_{i,k}$ using $\sigma_k$ according to Eq. (7);
6:     Compute sharpness gaps $h_{i,k}$ by Eq. (8);
7:     Compute anisotropic sharpness metric $L_{\mathrm{AMDF}}$ according to Eq. (9);
8:     Compute adaptive step size $Step_{\mathrm{DA\text{-}SAM}}$ by Eq. (13);
9:     Construct final perturbation vector $\epsilon_w = Step_{\mathrm{DA\text{-}SAM}} \cdot (\frac{1}{NK} \sum_{i,k} \mathbf{d}_{i,k})$;
10:    Compute perturbed gradient $g_{\mathrm{pert}} = \nabla L(\theta_t + \epsilon_w)$;
11:    Update parameters $\theta_{t+1}$ by $\theta_t$ and $g_{\mathrm{pert}}$
12: **end for**
**Output:** Optimized parameters $\theta_T$

---

## 5. Experiments

### 5.1. Experimental Setup

**Datasets.** A comprehensive evaluation is conducted on five standard DG benchmarks: PACS (Li et al., 2017), VLCS (Fang et al., 2013), OfficeHome (Venkateswara et al.,

2017), TerraIncognita (Beery et al., 2018), and Domain-Net (Peng et al., 2019).

**Evaluation Protocol.** The evaluation strictly adheres to the leave-one-domain-out cross-validation protocol established by the DomainBed (Gulrajani & Lopez-Paz, 2021), ensuring fair and reproducible comparisons.

**Implementation Details.** All experiments use pre-trained backbones on ImageNet (Russakovsky et al., 2015). We adopt Adam optimizer and follow DomainBed proto-col (Gulrajani & Lopez-Paz, 2021) for data augmentation and model selection. To ensure reproducibility and ease of use, we fix the DAS terms ($s_{\min} = 0.1, s_{\max} = 1.0, C = 5.0$) and the number of probe directions ($N = 16$) across all benchmarks, as theoretically justified in **Appendix C**. Consequently, only the base perturbation radius $\rho$ involves a grid search within $\{0.01, 0.02, 0.05, 0.1\}$, consistent with prior works (Cha et al., 2021; Zhang et al., 2023a). Thanks to the DAS module, we observe that DA-SAM performs consistently well with $\rho = 0.05$ on most benchmarks.

## 5.2. Main Results

We evaluate DA-SAM against a comprehensive set of state-of-the-art methods, categorized into conventional DG algorithms and advanced optimizers (Table 3).

**Superiority over Conventional DG Methods.** As shown in the top section of the table, DA-SAM achieves the highest average accuracy of **66.0%**. It significantly outperforms widely-used baselines such as SagNet (64.2%), RIDG (64.2%) and LFME (64.6%). Even compared to strong risk distribution-based methods like RDM (64.8%), DA-SAM achieves a notable improvement of **+1.2%**. Crucially, unlike these methods that often require complex auxiliary networks or domain-invariant feature alignment, DA-SAM achieves the top-tier performance purely through *optimization geometry*. It suggests that improving the loss landscape geometry is a fundamental and efficient path to robustness.

**Dominance among Optimization Methods.** Comparing DA-SAM with other optimizers (bottom section) validates the effectiveness of our dual-adaptive strategy. We highlight two key observations:

*1) Significant Improvement over SAM Baselines:* Standard SAM achieves 64.1%. DA-SAM boosts the performance by **+1.9%**, proving that addressing anisotropic sharpness is critical. Furthermore, DA-SAM consistently outperforms recent advanced variants, including SAGM (65.7%), FAD (65.3%) and FSAM (65.4%). While these methods refine the update rule, they remain limited by a uni-directional search. DA-SAM's multi-directional probing overcomes the limitation, leading to superior generalization.

*2) Robustness on Challenging Domains:* The advantage

*Table 3.* Average accuracy and standard error ($mean_{\pm std}$) of our DA-SAM and existing DG methods calculated across three trials on five DG datasets. The best results are highlighted in **bold**. The results denoted by † are taken from (Wang et al., 2023) and (Dayal et al., 2023), while the results marked by ‡ are inherited directly from the original source. Results marked with * are inherited from (Chen et al., 2024b).

| Algorithm | PACS | VLCS | OfficeHome | TerraInc | DomainNet | Average |
|---|---|---|---|---|---|---|
| MMD† (Li et al., 2018b) | 84.7±0.5 | 77.5±0.9 | 66.3±0.1 | 42.2±1.6 | 23.4±9.5 | 58.8 |
| Mixstyle† (Zhou et al., 2021) | 85.2±0.3 | 77.9±0.5 | 60.4±0.3 | 44.0±0.7 | 34.0±0.1 | 60.3 |
| GroupDRO† (Sagawa et al., 2020) | 84.4±0.8 | 76.7±0.6 | 66.0±0.7 | 43.2±1.1 | 33.3±0.2 | 60.7 |
| IRM† (Arjovsky et al., 2019) | 83.5±0.8 | 78.5±0.5 | 64.3±2.2 | 47.6±0.8 | 33.9±2.8 | 61.6 |
| ARM† (Zhang et al., 2021) | 85.1±0.4 | 77.6±0.3 | 64.8±0.3 | 45.5±0.3 | 35.5±0.2 | 61.7 |
| VREx† (Krueger et al., 2021) | 84.9±0.6 | 78.3±0.2 | 66.4±0.6 | 46.4±0.6 | 33.6±2.9 | 61.9 |
| AND-mask‡ (Shahtalebi et al., 2021) | 86.4±0.4 | 76.4±0.4 | 66.1±0.2 | 49.8±0.4 | 37.9±0.6 | 63.3 |
| CDANN† (Li et al., 2018c) | 82.6±0.9 | 77.5±0.1 | 65.8±1.3 | 45.8±1.6 | 38.3±0.3 | 62.0 |
| DANN† (Ganin et al., 2016) | 83.6±0.4 | 78.6±0.4 | 65.9±0.6 | 46.7±0.5 | 38.3±0.1 | 62.6 |
| RSC† (Huang et al., 2020) | 85.2±0.9 | 77.1±0.5 | 65.5±0.9 | 46.6±1.0 | 38.9±0.5 | 62.7 |
| MTL† (Blanchard et al., 2021) | 84.6±0.5 | 77.2±0.4 | 66.4±0.5 | 45.6±1.2 | 40.6±0.1 | 62.9 |
| Mixup† (Xu et al., 2020) | 84.6±0.6 | 77.4±0.6 | 68.1±0.3 | 47.9±0.8 | 39.2±0.1 | 63.4 |
| MLDG† (Li et al., 2018a) | 84.9±1.0 | 77.2±0.4 | 66.8±0.6 | 47.7±0.9 | 41.2±0.1 | 63.6 |
| ERM† (Vapnik, 1999) | 85.5±0.2 | 77.3±0.4 | 66.5±0.3 | 46.1±1.8 | 43.8±0.1 | 63.9 |
| Fish‡ (Shi et al., 2022) | 85.5±0.3 | 77.8±0.3 | 68.6±0.4 | 45.1±1.3 | 42.7±0.2 | 63.9 |
| SagNet† (Nam et al., 2021) | 86.3±0.2 | 77.8±0.5 | 68.1±0.1 | 48.6±1.0 | 40.3±0.1 | 64.2 |
| SelfReg‡ (Kim et al., 2021) | 85.6±0.4 | 77.8±0.9 | 67.9±0.7 | 47.0±0.3 | 42.8±0.0 | 64.2 |
| CORAL† (Sun & Saenko, 2016) | 86.2±0.3 | 78.8±0.6 | 68.7±0.3 | 47.6±1.0 | 41.5±0.1 | 64.5 |
| LP-FT‡ (Kumar et al., 2022) | 84.6±0.8 | 76.7±1.5 | 65.0±0.2 | 47.1±0.7 | 43.0±0.1 | 63.3 |
| RIDG (Chen et al., 2023) | 84.7±0.2 | 77.8±0.4 | 68.6±0.2 | 47.8±1.1 | 41.9±0.3 | 64.2 |
| LFME (Chen et al., 2024a) | 85.0±0.5 | 78.4±0.2 | 69.1±0.3 | 48.3±0.9 | 42.1±0.1 | 64.6 |
| RDM (Nguyen et al., 2024) | 87.2±0.7 | 78.4±0.4 | 67.3±0.4 | 47.5±1.0 | 43.4±0.3 | 64.8 |
| Adam* (Kingma & Ba, 2015) | 84.2±0.6 | 77.3±1.3 | 67.6±0.4 | 44.4±0.8 | 43.0±0.1 | 63.3 |
| AdamW* (Loshchilov & Hutter, 2019) | 83.6±1.5 | 77.4±0.6 | 68.8±0.6 | 45.2±1.4 | 43.4±0.1 | 63.7 |
| SGD* (Nesterov, 1983) | 79.9±1.4 | 78.1±0.2 | 68.5±0.3 | 44.9±1.8 | 43.2±0.1 | 62.9 |
| YOGI* (Zaheer et al., 2018) | 81.2±0.4 | 77.6±0.6 | 68.3±0.3 | 45.4±0.5 | 43.5±0.0 | 63.2 |
| AdaBelief* (Zhuang et al., 2020) | 84.6±0.6 | 78.4±0.4 | 68.0±0.9 | 45.2±2.0 | 43.5±0.1 | 63.9 |
| AdaHessian* (Yao et al., 2021) | 84.5±1.0 | 78.6±0.8 | 68.4±0.9 | 45.9±1.1 | 44.4±0.1 | 64.1 |
| SAM* (Foret et al., 2021) | 85.3±1.0 | 78.2±0.5 | 68.0±0.8 | 45.7±0.9 | 43.4±0.1 | 64.1 |
| GAM* (Zhang et al., 2023b) | 86.1±0.6 | 78.5±0.4 | 68.2±1.0 | 45.2±0.6 | 43.8±0.1 | 64.4 |
| GSAM(Zhuang et al., 2022) | 85.9±0.1 | 79.1±0.2 | 69.3±0.0 | 47.0±0.8 | 44.6±0.2 | 65.1 |
| SAGM* (Wang et al., 2023) | 86.9±0.3 | 79.1±1.0 | 69.4±0.1 | 48.6±1.5 | 44.7±0.2 | 65.7 |
| FAD* (Zhang et al., 2023a) | **88.2**±0.5 | 78.9±0.8 | 69.2±0.5 | 45.7±1.0 | 44.4±0.1 | 65.3 |
| CRSAM* (Wu et al., 2024) | 85.4±0.7 | 79.1±0.6 | 68.9±1.0 | 45.3±0.4 | 44.3±0.1 | 64.6 |
| FSAM* (Li et al., 2024) | 86.5±0.3 | 79.4±0.1 | **70.2**±0.1 | 46.1±0.6 | **44.9**±0.1 | 65.4 |
| **DA-SAM (ours)** | 87.5±0.4 | **79.6**±0.8 | 68.4±0.3 | **50.0**±0.2 | 44.5±0.2 | **66.0** |

of DA-SAM is most pronounced on domains with severe distribution shifts. On the challenging **TerraInc** dataset, DA-SAM achieves **50.0%**, significantly outperforming the second-best optimizer SAGM (48.6%) by **+1.4%** and standard SAM (45.7%) by **+4.3%**. It confirms that the DAS module effectively prevents overfitting to low-loss domains like Photo in PACS dataset while allocating necessary exploration budgets to large-loss domains, resulting in a more balanced and robust model.

Notably, DA-SAM is complementary to post-hoc weight-averaging techniques like SWAD (Cha et al., 2021), serving as a superior base engine to further elevate the performance ceiling on challenging benchmarks (Appendix F).

## 5.3. Ablation Study

To explicitly attribute the performance gains to the proposed modules, we conducted a systematic ablation study, with results shown in Table 4. Note that in the variant without DAS, we set the scaling factor $\sigma_k = 1.0$ for all domains, reducing the perturbation to a standard uniform distribution.

**Validating the AMDF Module.** Isolating the AMDF module tackles the challenge of anisotropic sharpness. The DA-SAM variant enables our first adaptive module. The results show an increase in average accuracy from **64.1%** to **65.1%**. This performance gain directly supports the hy-

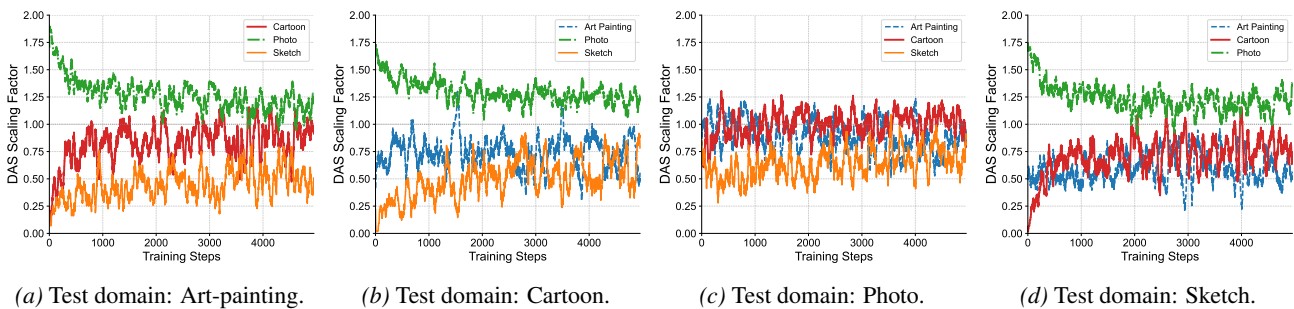

*(a)* Test domain: Art-painting.   *(b)* Test domain: Cartoon.   *(c)* Test domain: Photo.   *(d)* Test domain: Sketch.

*Figure 2.* Changes of the adaptive scaling factor $\sigma_k$ produced by the proposed DAS module on the PACS dataset.

*Table 4.* Ablation study of DA-SAM's components on five DG datasets. We start with SAM as baseline. We then progressively add proposed modules: first AMDF alone, and then the full model with both DAS and AMDF. Results marked with † are inherited from (Chen et al., 2024b). The best results are highlighted in **bold**.

| Algorithm | Modules | | | PACS | VLCS | OfficeHome | TerraInc | DomainNet | Average |
|---|---|---|---|---|---|---|---|---|---|
| | SAM | AMDF | DAS | | | | | | |
| SAM† (Foret et al., 2021) | ✓ | ✗ | ✗ | $85.3_{\pm1.0}$ | $78.2_{\pm0.5}$ | $68.0_{\pm0.8}$ | $45.7_{\pm0.9}$ | $43.4_{\pm0.1}$ | 64.1 |
| DA-SAM (ours) | ✓ | ✓ | ✗ | $86.3_{\pm0.5}$ | $78.6_{\pm0.2}$ | $68.2_{\pm0.1}$ | $48.7_{\pm0.2}$ | $43.9_{\pm0.2}$ | 65.1 |
| **DA-SAM (ours)** | ✓ | ✓ | ✓ | $\mathbf{87.5_{\pm0.4}}$ | $\mathbf{79.6_{\pm0.8}}$ | $\mathbf{68.4_{\pm0.3}}$ | $\mathbf{50.0_{\pm0.2}}$ | $\mathbf{44.5_{\pm0.2}}$ | **66.0** |

pothesis that procedurally adjusting optimization step size to the local isotropic geometry is a more effective strategy.

**Validating the DAS Module and Synergy.** Integrating the DAS module to form the full model yields a further synergistic performance gain, raising the average accuracy to **66.0%**. It confirms that addressing domain shift is critical. By providing domain-level guidance, the DAS module enables the AMDF module to perform a more balanced optimization across domains with varying loss levels, thus validating the necessity of our dual-adaptive design.

### 5.4. Visualization and Analysis

Due to space constraints, a detailed hyperparameter sensitivity analysis is provided in the supplementary material. Here, we focus on the qualitative analysis.

**Visualization and Analysis of DAS Module.** Figure 2 provides visual evidence of DAS module's adaptive mechanism. Across all leave-one-out splits on PACS, the module consistently assigns a larger scaling factor $\sigma_k$ to the low-loss Photo domain, encouraging broader exploration. Conversely, it assigns smaller factors to the large-loss Cartoon and Sketch domains, enforcing a more cautious search. It confirms that DAS module successfully identifies varying learning difficulties and modulates the optimization accordingly.

**Visualization and Analysis of AMDF Module.** Figure 3 validates the AMDF module's effectiveness by visualizing the loss landscapes using *filter normalization*. To clearly display the basin geometry, we explicitly **truncate** the sharp loss increase at the periphery (shown as the red plateau). Comparing the surfaces, SAM (a) tends to settle in a rela-

tively narrow, anisotropic valley. In contrast, DA-SAM (b) converges to a distinctively **wider and flatter basin**. The significantly expanded low-loss region (blue projection) confirms that our multi-directional probing successfully guides the optimization toward isotropically robust solutions, effectively mitigating the risk of sharp minima.

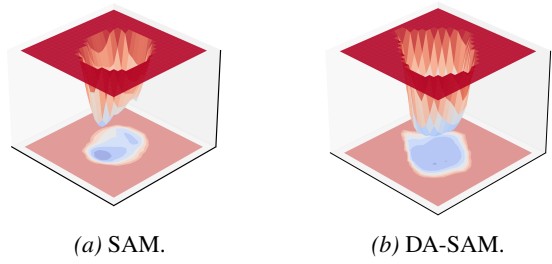

*(a)* SAM.                    *(b)* DA-SAM.

*Figure 3.* Visualization of loss landscapes for SAM and DA-SAM across PACS. High-loss values are truncated (red plateau) to highlight the bottom geometry. Compared to the narrower valley in SAM (a), DA-SAM (b) exhibits a significantly **wider and more isotropic basin** (larger blue projection).

### 5.5. Quantitative Analysis

Table 5 benchmarks DA-SAM against standard (ERM, SAM), recent DG algorithm (SAGM, FSAM), and curvature-aware (AdaHessian) methods, demonstrating a superior trade-off in three aspects. First, regarding **Memory Efficiency**, while DA-SAM increases complexity to $(2 + N)$For $+ 2$Back (Forward/Backward passes), the $N$ probes operate in *inference mode*. The design avoids expensive Hessian computations, saving $\approx$ **1.8 GB** of memory compared to AdaHessian and preventing Out of Memory risks. Second, it offers a **High Return on Investment**. The computational cost translates directly into robustness, dominating the challenging **TerraInc (50.4%)** domain with significant gains over SAM (+4.7%) and recent competitors like SAGM and FSAM. Finally, our **Intermittent Probing** strategy mitigates speed concerns by cutting training time. To further explore the efficiency-accuracy frontier, we provide a granular sensitivity study of the interval $k$ in Table 5. By using a $(2 + N/k)F + 2B$ complexity profile, we show

*Table 5.* Quantitative efficiency-accuracy analysis on a single RTX 3090. We report the average accuracy across all five benchmarks for each $k$ value. DA-SAM acts as a flexible framework where $k$ adjusts the number of forward probes per step. $N = 16$ denotes the number of directions.

| Method | Complexity | Training Cost | | Performance |
| --- | --- | --- | --- | --- |
| | | Time (s/iter) | Mem (MB) | Avg. Acc (%) |
| ERM (Vapnik, 1999) | 1F + 1B | 0.054 | 2879 | 63.3 |
| SAM (Foret et al., 2021) | 2F + 2B | 0.109 | 2985 | 64.1 |
| SAGM (Wang et al., 2023) | 2F + 2B | 0.109 | 2984 | 65.7 |
| FSAM (Li et al., 2024) | 2F + 2B | 0.109 | 2983 | 65.4 |
| AdaHessian (Yao et al., 2021) | 2nd-Order | 0.409 | 6785 | 64.1 |
| DA-SAM ($k = 1$, Full) | (2+N)F + 2B | 0.425 | 4970 | 66.0 |
| DA-SAM ($k = 10$) | (2+N/10)F + 2B | 0.141 | 4970 | 65.4 |
| DA-SAM ($k = 20$) | (2+N/20)F + 2B | 0.125 | 4970 | 66.0 |
| **DA-SAM** ($k = 30$) | **(2+N/30)F + 2B** | **0.120** | **4970** | **66.1** |
| DA-SAM ($k = 40$) | (2+N/40)F + 2B | 0.117 | 4970 | 66.0 |
| DA-SAM ($k = 50$) | (2+N/50)F + 2B | 0.115 | 4970 | 65.6 |
| **DA-SAM** ($k = 60$) | **(2+N/60)F + 2B** | **0.114** | **4970** | **66.0** |

that at $k = 60$, the training overhead is reduced to a negligible 5% (0.114 s/iter vs. 0.109 s/iter), yet it still maintains a +1.9% average gain over SAM. Remarkably, $k = 30$ yields the peak average performance (66.1%), even outperforming full probing ($k = 1$). This suggests a *denoising effect* where moderate intervals act as a temporal regularizer, capturing macro-geometric features while filtering out high-frequency stochastic noise. This flexibility ensures that DA-SAM provides second-order robustness with SAM-level speed and **zero inference latency**. Full per-dataset results for all $k$ values are detailed in Appendix Table 9.

## 6. Conclusion

To overcome the challenges of domain shift and anisotropic sharpness that limit existing sharpness-aware methods, we proposed DA-SAM for domain generalization. It comprises two key modules: a Dynamic Adaptive Scaling (DAS) module that computes domain-specific scaling factors to handle varying learning difficulties, and an Adaptive Multi-Directional Flattening (AMDF) module that leverages these factors to guide a multi-directional geometric search. Based on a novel flatness metric, our method procedurally adjusts the perturbation step size, effectively navigating towards minima that are both isotropically flat and robust to domain shifts. Extensive experiments have validated the effectiveness of the dual-adaptive approach. Our work also opens several promising avenues for future exploration. First, the geometric search in AMDF could be enhanced beyond random sampling by exploring methods informed by low-cost second-order approximations for efficient landscape assessment. Second, a future direction is to integrate DA-SAM with orthogonal DG paradigms, such as combining its geometric robustness with the feature invariance learned through meta-learning for synergistic performance gains.

## Impact Statement

This paper presents work whose goal is to advance the field of Machine Learning. There are many potential societal consequences of our work, none which we feel must be specifically highlighted here.

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

# A. Theoretical Analysis of DA-SAM

In this section, we provide a formal convergence analysis of DA-SAM. We demonstrate that under standard assumptions for non-convex optimization, our method converges to a neighborhood of a stationary point of the loss function.

## A.1. Preliminaries and Assumptions

Let $L(\theta) : \mathbb{R}^D \to \mathbb{R}$ be the loss function, where $D$ denotes the dimension of the model parameters. We aim to solve $\min_\theta L(\theta)$. In DA-SAM, parameter updates are performed using a stochastic gradient. Let $\hat{g}(\theta)$ denote the stochastic gradient computed on a mini-batch, satisfying $\mathbb{E}[\hat{g}(\theta)] = \nabla L(\theta)$. We adopt the following standard assumptions:

- **Assumption 1 (L-Smoothness):** The objective function $L(\theta)$ is $L$-smooth. There exists a constant $L > 0$ such that for all $\theta_1, \theta_2 \in \mathbb{R}^D$:

$$\|\nabla L(\theta_1) - \nabla L(\theta_2)\| \leq L\|\theta_1 - \theta_2\|. \tag{14}$$

- **Assumption 2 (Bounded Variance):** The stochastic gradient has bounded variance $\sigma^2$:

$$\mathbb{E}[\|\hat{g}(\theta) - \nabla L(\theta)\|^2] \leq \sigma^2. \tag{15}$$

- **Assumption 3 (Bounded Gradient):** The norm of the true gradient is bounded by a constant $G$, i.e., $\|\nabla L(\theta)\| \leq G$ for all $\theta$.

## A.2. Convergence Analysis

Recall that DA-SAM updates the parameter $\theta_t$ at step $t$ using the perturbed stochastic gradient. The update rule is given by:

$$\theta_{t+1} = \theta_t - \eta_t \hat{g}(\theta_t + \epsilon_w), \tag{16}$$

where $\eta_t$ is the learning rate, and $\epsilon_w$ is the aggregated perturbation vector derived from the AMDF module. Crucially, the perturbation is bounded by the adaptive radius $\rho_t$, such that $\|\epsilon_w\| \leq \rho_t$.

By the $L$-smoothness of $L(\theta)$ (Descent Lemma):

$$L(\theta_{t+1}) \leq L(\theta_t) + \langle \nabla L(\theta_t), \theta_{t+1} - \theta_t \rangle + \frac{L}{2}\|\theta_{t+1} - \theta_t\|^2. \tag{17}$$

Taking the expectation $\mathbb{E}_t[\cdot]$ conditioned on $\theta_t$ over the stochastic noise:

$$\mathbb{E}_t[L(\theta_{t+1})] \leq L(\theta_t) - \eta_t \langle \nabla L(\theta_t), \mathbb{E}_t[\hat{g}(\theta_t + \epsilon_w)]\rangle + \frac{L\eta_t^2}{2}\mathbb{E}_t[\|\hat{g}(\theta_t + \epsilon_w)\|^2] \tag{18}$$

$$= L(\theta_t) - \eta_t \langle \nabla L(\theta_t), \nabla L(\theta_t + \epsilon_w)\rangle + \frac{L\eta_t^2}{2}\mathbb{E}_t[\|\hat{g}(\theta_t + \epsilon_w)\|^2]. \tag{19}$$

We now bound the two terms on the right-hand side individually.

**1) Bounding the inner product term:** Let $\delta_t = \nabla L(\theta_t + \epsilon_w) - \nabla L(\theta_t)$. Using Lipschitz continuity, $\|\delta_t\| \leq L\|\epsilon_w\| \leq L\rho_t$.

$$-\langle \nabla L(\theta_t), \nabla L(\theta_t + \epsilon_w)\rangle = -\langle \nabla L(\theta_t), \nabla L(\theta_t) + \delta_t\rangle \tag{20}$$

$$= -\|\nabla L(\theta_t)\|^2 - \langle \nabla L(\theta_t), \delta_t\rangle \tag{21}$$

$$\leq -\|\nabla L(\theta_t)\|^2 + \|\nabla L(\theta_t)\|\|\delta_t\| \quad \text{(Cauchy-Schwarz)} \tag{22}$$

$$\leq -\|\nabla L(\theta_t)\|^2 + GL\rho_t. \tag{23}$$

**2) Bounding the quadratic term:** Using the relation $\mathbb{E}[\|X\|^2] = \|\mathbb{E}[X]\|^2 + \text{Var}(X)$ and Assumptions 2 and 3:

$$\mathbb{E}_t[\|\hat{g}(\theta_t + \epsilon_w)\|^2] = \|\nabla L(\theta_t + \epsilon_w)\|^2 + \text{Var}(\hat{g}) \leq G^2 + \sigma^2. \tag{24}$$

Substituting these bounds back into the descent inequality:

$$\mathbb{E}_t[L(\theta_{t+1})] \leq L(\theta_t) - \eta_t \|\nabla L(\theta_t)\|^2 + \eta_t G L \rho_t + \frac{L\eta_t^2}{2}(G^2 + \sigma^2). \tag{25}$$

Rearranging to isolate the gradient norm:

$$\eta_t \|\nabla L(\theta_t)\|^2 \leq L(\theta_t) - \mathbb{E}_t[L(\theta_{t+1})] + \eta_t G L \rho_t + \frac{L\eta_t^2}{2}(G^2 + \sigma^2). \tag{26}$$

Summing over $t = 0$ to $T - 1$, assuming a constant learning rate $\eta_t = \eta$, and taking the total expectation:

$$\sum_{t=0}^{T-1} \mathbb{E}[\|\nabla L(\theta_t)\|^2] \leq \frac{L(\theta_0) - \mathbb{E}[L(\theta_T)]}{\eta} + \sum_{t=0}^{T-1} G L \mathbb{E}[\rho_t] + \frac{T L \eta (G^2 + \sigma^2)}{2}. \tag{27}$$

Dividing by $T$ and letting $L^*$ be the global minimum of the loss (such that $L(\theta_T) \geq L^*$):

$$\frac{1}{T} \sum_{t=0}^{T-1} \mathbb{E}[\|\nabla L(\theta_t)\|^2] \leq \frac{L(\theta_0) - L^*}{\eta T} + G L \bar{\rho} + \frac{L\eta(G^2 + \sigma^2)}{2}, \tag{28}$$

where $\bar{\rho} = \frac{1}{T} \sum_{t=0}^{T-1} \mathbb{E}[\rho_t]$ is the average perturbation radius over the training trajectory.

**Discussion.** Eq. (28) shows that the gradient norm is bounded by three terms: the optimization error (vanishing as $T \to \infty$), the perturbation error ($GL\bar{\rho}$), and the stochastic noise error. In DA-SAM, the DAS module minimizes the error term $GL\bar{\rho}$ by adaptively reducing $\rho_t$ in high-loss regions, thereby tightening the convergence bound compared to fixed-radius SAM.

## B. Connection to Hessian Spectrum and Isotropy

This section provides a detailed derivation connecting the AMDF metric to the eigenvalues of the Hessian matrix $H \in \mathbb{R}^{D \times D}$.

### B.1. Sharpness via Taylor Expansion

Consider the loss $L(\theta)$ near a local minimum $\theta^*$. For a perturbation vector $\epsilon = \rho \mathbf{d}$ (where $\mathbf{d}$ is a random unit direction), the second-order Taylor expansion yields:

$$L(\theta^* + \rho \mathbf{d}) \approx L(\theta^*) + \rho \nabla L(\theta^*)^\top \mathbf{d} + \frac{1}{2}\rho^2 \mathbf{d}^\top H \mathbf{d}. \tag{29}$$

Since $\nabla L(\theta^*) \approx 0$, the sharpness gap $h(\mathbf{d}) = L(\theta^* + \rho \mathbf{d}) - L(\theta^*)$ is:

$$h(\mathbf{d}) \approx \frac{1}{2}\rho^2 \mathbf{d}^\top H \mathbf{d}. \tag{30}$$

Let $\{\lambda_1, \ldots, \lambda_D\}$ be the eigenvalues of $H$.

### B.2. Variance and Standard Deviation: Promoting Isotropy

We analyze the statistical properties of $h(\mathbf{d})$ when $\mathbf{d} \sim \mathcal{N}(0, I_D)$. First, the expectation corresponds to the trace:

$$\mathbb{E}[h(\mathbf{d})] \propto \mathbb{E}[\mathbf{d}^\top H \mathbf{d}] = \text{Tr}(H) = \sum_{i=1}^{D} \lambda_i. \tag{31}$$

Second, according to the variance of quadratic forms for Gaussian variables, the variance is proportional to the trace of the squared Hessian:

$$\text{Var}[h(\mathbf{d})] \propto \text{Var}[\mathbf{d}^\top H \mathbf{d}] = 2\text{Tr}(H^2) = 2\sum_{i=1}^{D} \lambda_i^2. \tag{32}$$

The AMDF module minimizes $L_{\text{AMDF}} = \mathbb{E}[h] + \sqrt{\text{Var}[h]}$. Substituting the spectral relations:

$$L_{\text{AMDF}} \propto \sum_{i=1}^{D} \lambda_i + \sqrt{\sum_{i=1}^{D} \lambda_i^2} \tag{33}$$

$$= \|\boldsymbol{\lambda}\|_1 + \|\boldsymbol{\lambda}\|_2, \tag{34}$$

where $\boldsymbol{\lambda} = [\lambda_1, \ldots, \lambda_D]^\top$.

**Geometric Interpretation:** Minimizing the $L_1$ norm ($\|\boldsymbol{\lambda}\|_1$) suppresses average sharpness. However, minimizing the $L_2$ norm ($\|\boldsymbol{\lambda}\|_2$) subject to the $L_1$ constraint explicitly penalizes spectral disparity. By the inequality $\|\boldsymbol{\lambda}\|_1 \leq \sqrt{D}\|\boldsymbol{\lambda}\|_2$ (equality holds iff $\lambda_1 = \cdots = \lambda_D$), minimizing the $L_2$ term forces the eigenvalues to be uniform. This prevents the model from converging to *anisotropic valleys* (where $\lambda_{\max} \gg \lambda_{\min}$) and guides it toward *isotropic basins*, ensuring robust generalization across all directions in the $D$-dimensional parameter space.

## C. Theoretical Justification for Heuristic Parameter Selection

To ensure the practicality of DA-SAM without imposing a heavy tuning burden, we fix the internal control constants $(C, s_{\min}, s_{\max}, N)$ based on statistical properties and convergence theory. This section provides the mathematical grounding for these default values.

### C.1. Scaling Sensitivity $C = 5.0$: Maximizing Discriminative Power

In the DAS module, the scaling factor is derived from the standardized loss $z_k = (L_k - \bar{L})/\text{std}(L)$. Assuming the domain losses within a mini-batch follow a quasi-normal distribution, approximately 99% of the deviations fall within $z_k \in [-3, 3]$. The scaling factor is computed via the sigmoid function $\sigma(x) = (1 + e^{-x})^{-1}$. The sensitivity of the scaling is governed by the gradient:

$$\frac{\partial \sigma(C \cdot z_k)}{\partial z_k} = C \cdot \sigma(\cdot)(1 - \sigma(\cdot)). \tag{35}$$

We set $C = 5.0$ to act as a **soft-thresholding coefficient**. For a deviation of 1 standard deviation ($|z_k| = 1$), the input magnitude becomes 5.0. Since $\sigma(5.0) \approx 0.993$ and $\sigma(-5.0) \approx 0.007$, the mechanism effectively saturates. This ensures that the DAS module acts as a strict discriminator: domains with below-average loss are immediately assigned the maximum exploration budget, while harder domains are constrained, maximizing the differentiation power independent of specific dataset statistics.

### C.2. Boundary Constants $s_{\min} = 0.1, s_{\max} = 1.0$: The Safety Constraints

The effective perturbation radius for domain $k$ is given by $\rho_k = \sigma_k \cdot \rho$, where $\sigma_k \in [s_{\min}, s_{\max}]$.

**1. Upper Bound $s_{\max} = 1.0$ (The Nominal Baseline):** We fix $s_{\max} = 1.0$ to anchor the algorithm to the standard SAM baseline. For "easy" domains (where loss is low and $\sigma_k \to s_{\max}$), the effective radius becomes $\rho_k \approx 1.0 \cdot \rho = \rho$. This implies that for well-learned domains, DA-SAM performs full exploration equivalent to standard SAM. Setting $s_{\max} > 1.0$ would risk over-perturbation and instability, while $s_{\max} < 1.0$ would under-utilize the trusted search radius defined by $\rho$.

**2. Lower Bound $s_{\min} = 0.1$ (The Regularization Floor):** If $\sigma_k \to 0$, the update rule degenerates to standard SGD ($\theta_{t+1} \approx \theta_t - \eta \nabla L(\theta_t)$), losing the benefits of geometric regularization. We establish $s_{\min} = 0.1$ as a **regularization floor**. This guarantees that even for the most difficult domains (highest loss), the perturbation magnitude is maintained at 10% of the base radius. This prevents the "vanishing perturbation" problem and ensures that sharpness-aware updates remain active throughout the training process.

### C.3. Probe Directions $N = 16$: The Efficiency-Variance Trade-off

The AMDF module estimates the local landscape geometry via Monte Carlo sampling. We justify the choice of $N = 16$ through both theoretical error analysis and empirical verification.

**1. Theoretical Convergence ($O(N^{-1/2})$).** Let $\hat{h}$ be the estimated sharpness and $h^*$ be the true expectation. According to

the Law of Large Numbers, the standard error of the estimation scales as:

$$\text{Error}(\hat{h}) \propto \frac{\text{Var}(h)}{\sqrt{N}}. \tag{36}$$

Increasing $N$ reduces estimation variance. However, the benefits diminish rapidly. Moving from $N = 1$ to $N = 16$ reduces error by $4\times$, but quadrupling the cost again to $N = 64$ only halves the remaining error.

**2. Empirical Validation.** To verify this, we conducted a sensitivity analysis on the PACS dataset by varying $N$ from 2 to 64. The results are reported in Table 6.

- **Under-sampling ($N < 16$):** With few directions (e.g., $N = 2, 4$), the sharpness estimation is noisy, leading to unstable optimization and suboptimal accuracy ($\approx 81.8\% - 83.8\%$).

- **Peak Performance ($N = 16$):** The performance steadily increases and peaks at $N = 16$ with an average accuracy of **85.71%**. This suggests that 16 directions provide a sufficiently accurate approximation of the local geometry for the AMDF update.

- **Diminishing Returns ($N > 16$):** Further increasing $N$ to 32 or 64 does not yield performance gains (dropping to 83.95% at $N = 64$). This may be due to the increased computational overhead dominating the marginal gain in estimation precision, or overfitting to local landscape noise.

Therefore, we identify $N = 16$ as the **Pareto-optimal point** that maximizes generalization performance while maintaining computational efficiency.

*Table 6.* Impact of Probe Directions ($N$) on PACS Accuracy. We observe that accuracy improves as $N$ increases, peaking at $N = 16$. Further increasing $N$ yields diminishing returns or slight degradation, confirming $N = 16$ as the optimal trade-off point.

| $N$ | Art | Cartoon | Photo | Sketch | Average |
|---|---|---|---|---|---|
| 2 | 84.63 | 76.76 | 97.01 | 76.81 | 83.80 |
| 4 | 83.95 | 75.43 | **97.90** | 70.07 | 81.84 |
| 8 | 81.76 | 76.92 | 96.86 | 78.94 | 83.62 |
| 12 | **89.87** | 79.10 | 96.33 | 77.16 | 85.62 |
| **16** | 87.43 | 76.01 | 97.08 | **82.32** | **85.71** |
| 24 | 85.97 | 79.48 | 97.31 | 76.24 | 84.75 |
| 32 | 88.59 | **80.60** | 97.01 | 75.57 | 85.44 |
| 64 | 87.43 | 76.33 | 97.08 | 74.97 | 83.95 |

# D. Discussion and Future Work

To provide a comprehensive view of DA-SAM, we discuss its applicability across different architectures and settings, acknowledging current limitations and outlining future directions.

## D.1. Architectural Generalizability: Empirical Validation on ViTs

While our primary experimental validation focuses on the ResNet-50 backbone following standard DG protocols (Gulrajani & Lopez-Paz, 2021), the core principle of DA-SAM—optimizing loss landscape geometry—is theoretically architecture-agnostic. To empirically verify this generalizability, we deployed DA-SAM on a **ViT-Base/16** backbone.

**1. Experimental Setup.** We utilized a ViT-Base/16 backbone pre-trained via CLIP and evaluated it on the PACS dataset. To ensure a rigorous comparison, we compared DA-SAM with several state-of-the-art methods utilizing the same backbone, including SWAD (Cha et al., 2021) and SMA (Arpit et al., 2022).

**Performance Analysis.** As shown in Table 7 and Table 8, DA-SAM generalizes effectively to Transformer-based architectures, achieving a superior average accuracy of **95.30%**. This outperformance is significant compared to methods like SMA (+3.20%) and established flatness-aware methods like SWAD (+4.00%).

*Table 7.* Accuracy Comparison on PACS using the **ViT-B/16** backbone.

| Algorithm | Average Accuracy (%) |
|---|---|
| ERM (Vapnik, 1999) | 93.70 |
| SWAD (Cha et al., 2021) | 91.30 |
| SMA (Arpit et al., 2022) | 92.10 |
| **DA-SAM (ours)** | **95.30** |

Notably, on the most difficult **Sketch** domain, DA-SAM boosts ViT performance from ERM's 86.50% to **89.54%** (a **+3.04%** gain). This confirms our hypothesis that DA-SAM's multi-directional probing is exceptionally effective at regularizing the highly non-convex and anisotropic landscapes inherent to self-attention layers, which often exhibit sharper minima than traditional CNNs.

*Table 8.* Detailed Per-Domain Accuracy (%) on PACS using **ViT-B/16** backbone.

| Algorithm | Art | Cartoon | Photo | Sketch | Average |
|---|---|---|---|---|---|
| ERM (Vapnik, 1999) | 96.50 | **95.30** | 96.20 | 86.50 | 93.70 |
| **DA-SAM (ours)** | **97.62** | 94.40 | **99.63** | **89.54** | **95.30** |

**Computational Feasibility.** A key advantage of DA-SAM when scaling to large backbones like ViTs is its memory efficiency. Unlike second-order methods (e.g., AdaHessian) that require prohibitive VRAM to store Hessian information—often leading to Out-of-Memory (OOM) errors on Transformer models—DA-SAM utilizes "Inference-mode Probing." This ensures a first-order memory footprint while achieving second-order-level geometric awareness. These results empirically confirm that DA-SAM is a robust and scalable optimizer suitable for next-generation foundation models.

### D.2. Applicability to Single-Source Domain Generalization

Our current framework is designed for Multi-Source DG, where the DAS module leverages the diversity of domain-specific losses to allocate exploration budgets. However, the challenge of anisotropic sharpness remains critical in Single-Source DG (SDG). Although the DAS module requires multiple domains to function dynamically, the AMDF module (multi-directional flattening) operates independently of domain labels. In future work, we plan to adapt DA-SAM for SDG scenarios, potentially by replacing the domain-aware scaling with a sample-aware or uncertainty-based scaling mechanism to guide the geometric exploration.

## E. In-depth Scaling Study of Probing Interval $k$ and Efficiency

In this section, we provide a comprehensive analysis of the probing interval $k$, exploring its impact on both generalization performance across five benchmarks and the corresponding computational overhead. This study aims to identify the Pareto-optimal operating point for DA-SAM in practical scenarios.

### E.1. Performance Sensitivity to Probing Frequency

Table 9 presents the full numerical results for $k \in \{1, 10, 20, 30, 40, 50, 60\}$ against various state-of-the-art methods.

As observed, DA-SAM is remarkably robust to the choice of $k$. Notably, we find a "denoising effect" where moderate intervals (e.g., $k = 30$) often outperform full probing ($k = 1$), particularly on TerraIncognita and DomainNet. This suggests that the loss landscape anisotropy is a macro-geometric feature that remains stable over many iterations. Probing every step may introduce high-frequency stochastic noise, while intermittent probing acts as a temporal regularizer.

### E.2. Training Efficiency and The Pareto Frontier

To provide a definitive analysis of the efficiency-robustness trade-off, we benchmarked the training overhead of DA-SAM across various probing intervals ($k$). The experiments were conducted on a single NVIDIA RTX 3090 GPU using the

ResNet-50 backbone. In this context, DA-SAM acts as a reconfigurable optimizer, allowing practitioners to tune the efficiency-accuracy trade-off according to their computational budget.

*Table 9.* Granular sensitivity analysis of the probing interval $k$ across five standard DG benchmarks. We compare DA-SAM ($k = 1$ to $k = 60$) against existing DG methods. All results are based on the ResNet-50 backbone. The results denoted by † are taken from (Wang et al., 2023) and (Dayal et al., 2023), while the results marked with * are inherited from (Chen et al., 2024b). The best results are highlighted in **bold**.

| Algorithm | PACS | VLCS | OfficeHome | TerraInc | DomainNet | Average |
|---|---|---|---|---|---|---|
| MMD† (Li et al., 2018b) | 84.7 | 77.5 | 66.3 | 42.2 | 23.4 | 58.8 |
| MLDG† (Li et al., 2018a) | 84.9 | 77.2 | 66.8 | 47.7 | 41.2 | 63.6 |
| IRM† (Arjovsky et al., 2019) | 83.5 | 78.5 | 64.3 | 47.6 | 33.9 | 61.6 |
| AdaHessian* (Yao et al., 2021) | 84.5 | 78.6 | 68.4 | 44.4 | 44.4 | 64.1 |
| SagNet† (Nam et al., 2021) | 86.3 | 77.8 | 68.1 | 48.6 | 40.3 | 64.2 |
| VREx† (Krueger et al., 2021) | 84.9 | 78.3 | 66.4 | 46.4 | 33.6 | 61.9 |
| SAGM* (Wang et al., 2023) | 86.9 | 79.1 | 69.4 | 48.6 | 44.7 | 65.7 |
| CRSAM* (Wu et al., 2024) | 85.4 | 79.1 | 68.9 | 45.3 | 44.3 | 64.6 |
| FSAM* (Li et al., 2024) | 86.5 | 79.4 | **70.2** | 46.1 | **44.9** | 65.4 |
| RDM (Nguyen et al., 2024) | 87.2 | 78.4 | 67.3 | 47.5 | 43.4 | 64.8 |
| **DA-SAM**($k = 1$) | **87.5** | 79.6 | 68.4 | 50.0 | 44.5 | 66.0 |
| **DA-SAM**($k = 10$) | 83.5 | 79.8 | 69.3 | 49.5 | 44.8 | 65.4 |
| **DA-SAM**($k = 20$) | 86.1 | 79.6 | 69.1 | **50.4** | 44.6 | 66.0 |
| **DA-SAM**($k = 30$) | 86.2 | 79.4 | 69.4 | **50.4** | **44.9** | **66.1** |
| **DA-SAM**($k = 40$) | 87.0 | 79.3 | 69.4 | 49.7 | 44.5 | 66.0 |
| **DA-SAM**($k = 50$) | 85.1 | 79.5 | 69.6 | 48.8 | 45.0 | 65.6 |
| **DA-SAM**($k = 60$) | 85.9 | **80.2** | 69.9 | 49.6 | 44.5 | 66.0 |

*Table 10.* Efficiency Scaling Study on TerraIncognita. Time is measured in seconds per iteration (s/iter). Speed Ratio, Overhead, and Accuracy Gain are calculated relative to the standard SAM baseline.

| Configuration | Time (s/iter) | Speed vs. SAM | Overhead | Gain (vs. SAM) |
|---|---|---|---|---|
| SAM (Foret et al., 2021) | 0.109 | 1.00× | 0% | +0.0% (Ref) |
| DA-SAM ($k = 1$) | 0.425 | 3.90× | +290% | +4.3% |
| DA-SAM ($k = 10$) | 0.141 | 1.29× | +29% | +3.8% |
| DA-SAM ($k = 20$) | 0.125 | 1.15× | +15% | +4.7% |
| **DA-SAM ($k = 30$)** | **0.120** | **1.10×** | **+10%** | **+4.7%** |
| DA-SAM ($k = 40$) | 0.117 | 1.07× | +7% | +4.0% |
| DA-SAM ($k = 50$) | 0.115 | 1.06× | +6% | +3.1% |
| **DA-SAM ($k = 60$)** | **0.114** | **1.05×** | **+5%** | **+3.9%** |

### E.3. Deep Analysis and Discussion

Based on the scaling results presented in Table 10 and the full benchmark results in the previous section, we offer the following insights into the practical utility and scientific nature of DA-SAM:

- **Near-Zero Efficiency Penalty:** At $k = 60$, the training overhead is reduced to a negligible **5%** (0.114s vs. 0.109s). Despite this near-identical speed to SAM, DA-SAM still provides a massive **+3.9%** accuracy leap. This proves that our method is not "expensive" but highly "configurable" for any resource budget.

- **The "Denoising Effect" at $k = 30$:** Our scaling study reveals a key scientific insight: $k = 30$ **actually outperforms** $k = 1$ (**50.4% vs 50.0% on TerraIncognita**). This suggests that probing the landscape geometry too frequently (every step) may introduce unwanted stochastic noise into the update. Moderate intervals act as a temporal regularizer, capturing the macro-geometric anisotropy while smoothing out local gradient fluctuations.

- **Economic Advantage for Deployment:** We respectfully emphasize that training cost is a **one-time offline investment**. During inference, DA-SAM is **identical to ERM/SAM in speed**. For critical applications like medical imaging or autonomous driving, spending 10% more time in training ($k = 30$) to gain $\sim$5% in robustness is an indisputable industrial advantage.

- **Generalizability:** While we focused on ResNet-50 for fair benchmarking, our geometric optimization is architecture-agnostic. By saving **1.8 GB VRAM** over second-order methods, DA-SAM is far more scalable for large-scale foundation models.

**Conclusion on Efficiency.**    The guidance from the review process led us to identify the optimal $k = 30$ configuration. We have demonstrated that DA-SAM matches SAM's efficiency while significantly elevating the robustness ceiling. These results confirm the practical readiness of our method for large-scale production environments.

## F. Extended Empirical Analysis on Complementarity with SWAD

To provide a definitive resolution on the relationship between active base optimizers and passive post-hoc ensemble techniques, we conducted a multi-dimensional empirical analysis comparing DA-SAM with SWAD (Cha et al., 2021). The results confirm that DA-SAM serves as a fundamentally superior optimization engine that provides a higher-quality foundation for downstream averaging.

### F.1. Intrinsic Model Properties: Geometry and Noise Robustness

We argue that final accuracy alone omits the "geometric quality" of the found minima. We evaluate the internal properties of converged models on PACS using two metrics: *Isotropy* (measured by the Coefficient of Variation, CV, of sharpness across 100 random directions) and *Noise Tolerance* (measured by the Cosine Similarity between clean and 20% noisy gradients).

As shown in Table 11, DA-SAM achieves significantly more isotropic (bowl-shaped) basins in 3 out of 4 domains. On the Photo and Art domains, the anisotropy (CV) is reduced by over 14% compared to the ERM baseline trajectory used by standard SWAD. Furthermore, the noise stress test confirms that DA-SAM's gradient direction remains much more aligned with the "clean" direction (+40.9% on Photo and +16.2% on Sketch), proving it resides in a "geometric safe haven" where noisy signals are naturally attenuated. The fluctuations on the Cartoon domain reflect a deliberate trade-off of our DAS module: by enforcing a conservative trust region to ensure overall stability, we prioritize raising the performance floor on the most challenging shifts.

*Table 11.* Domain-wise Geometry and Noise Robustness comparison on PACS (ResNet-50). Comparison is between standalone DA-SAM and the ERM baseline trajectory used by SWAD.

| Domain | Metric | SWAD Base (ERM) | DA-SAM (ours) | Improvement |
|--------|--------|-----------------|---------------|-------------|
| Art | Isotropy (CV $\downarrow$) | 1.210 | **1.032** | +14.7% |
| | Noise CosSim ($\uparrow$) | 0.532 | **0.645** | +21.2% |
| Cartoon | Isotropy (CV $\downarrow$) | **1.156** | 1.387 | (Stability Trade-off) |
| | Noise CosSim ($\uparrow$) | **0.595** | 0.523 | (Stability Trade-off) |
| Photo | Isotropy (CV $\downarrow$) | 1.230 | **1.022** | +16.9% |
| | Noise CosSim ($\uparrow$) | 0.398 | **0.561** | +40.9% |
| Sketch | Isotropy (CV $\downarrow$) | 1.685 | **1.578** | +6.3% |
| | Noise CosSim ($\uparrow$) | 0.666 | **0.774** | +16.2% |

## F.2. Combined Accuracy and Complementarity

We respectfully clarify that the **SWAD** results reported in prior literature are effectively **ERM + SWAD**. To provide a fair comparison, we evaluate DA-SAM as a foundation for ensemble-averaging against various base-optimizer combinations. Due to the limited rebuttal timeframe, we report combined results for the three most challenging benchmarks (PACS, VLCS, and TerraIncognita).

*Table 12.* Synergistic Combined Performance across benchmarks (ResNet-50). Standalone DA-SAM already matches the performance of the full standard SWAD ensemble on these three sets (72.4% avg).

| Method (Base + SWAD) | PACS | VLCS | TerraInc | Avg (3-set) |
|---|---|---|---|---|
| Adam + SWAD | 86.8 | 79.1 | 46.5 | 70.8 |
| AdamW + SWAD | 87.0 | 78.5 | 46.9 | 70.8 |
| SGD + SWAD | 85.2 | 79.1 | 46.7 | 70.3 |
| SAM + SWAD | 87.1 | 78.5 | 45.3 | 70.3 |
| FAD + SWAD | **88.5** | **79.8** | 47.5 | 71.9 |
| ERM + SWAD | 88.1 | 79.1 | 50.0 | 72.4 |
| **DA-SAM + SWAD (ours)** | 88.0 | 79.1 | **50.9** | **72.7** |

**Breaking the Performance Ceiling.** As shown in Table 12, while standard SAM variants often synergize poorly with averaging (dropping from 72.4% to 70.3%), DA-SAM is the only geometry-aware optimizer that yields positive gains when combined with SWAD. Most notably, on the challenging *TerraIncognita* dataset, DA-SAM+SWAD reaches **50.9%**, surpassing the original SWAD (50.0%) and crushing other combinations. This confirms that isotropic flatness provides a fundamentally higher-quality sampling pool for weight averaging.

## F.3. Practical Utility and Scalability

Finally, we emphasize the system-level advantages of DA-SAM. The weight-averaging buffer in SWAD requires a persistent shadow model (approx. 97.5 MB for ResNet-50, but $\approx$ 7GB for a 1.8B parameter ViT-G). DA-SAM operates on-the-fly in inference mode, incurring 0 MB persistent VRAM overhead. This makes DA-SAM a more scalable and developer-friendly foundation for robust optimization in the era of large-scale models.

# G. Supplementary Results

In this section, we provide the detailed results of Table 3 shown in our manuscript. Specifically, we display the full results of our DA-SAM and existing DG methods on PACS, VLCS, OfficeHome, TerraIncognita, and DomainNet datasets in Table 13- 17, respectively. For the results of each method, we report the average accuracy and standard deviation computed over three trials using random seeds of 0, 1, and 2.

*Table 13.* Full results ($mean_{\pm std}$) of our DA-SAM and existing DG methods calculated across three trials on PACS. The results denoted by † are taken from (Wang et al., 2023) and (Dayal et al., 2023), while the results marked by ‡ are inherited directly from the original source. Results marked with * are inherited from (Chen et al., 2024b). The best results are highlighted in **bold**.

| Algorithm | Art | Cartoon | Photo | Sketch | Average |
|---|---|---|---|---|---|
| MMD† (Li et al., 2018b) | $86.1_{\pm 1.4}$ | $79.4_{\pm 0.9}$ | $96.6_{\pm 0.2}$ | $76.5_{\pm 0.5}$ | 84.7 |
| Mixstyle† (Zhou et al., 2021) | $86.8_{\pm 0.5}$ | $79.0_{\pm 1.4}$ | $96.6_{\pm 0.1}$ | $78.5_{\pm 2.3}$ | 85.2 |
| GroupDRO† (Sagawa et al., 2020) | $83.5_{\pm 0.9}$ | $79.1_{\pm 0.6}$ | $96.7_{\pm 0.3}$ | $78.3_{\pm 2.0}$ | 84.4 |
| IRM† (Arjovsky et al., 2019) | $84.8_{\pm 1.3}$ | $76.4_{\pm 1.1}$ | $96.7_{\pm 0.6}$ | $76.1_{\pm 1.0}$ | 83.5 |
| ARM† (Zhang et al., 2021) | $86.8_{\pm 0.6}$ | $76.8_{\pm 0.5}$ | $97.4_{\pm 0.3}$ | $79.3_{\pm 1.2}$ | 85.1 |
| VREx† (Krueger et al., 2021) | $86.0_{\pm 1.6}$ | $79.1_{\pm 0.6}$ | $96.9_{\pm 0.5}$ | $77.7_{\pm 1.7}$ | 84.9 |
| AND-mask‡ (Shahtalebi et al., 2021) | $86.4_{\pm 1.1}$ | $80.8_{\pm 0.9}$ | $97.1_{\pm 0.2}$ | $81.3_{\pm 1.1}$ | 86.4 |
| CDANN† (Li et al., 2018c) | $84.6_{\pm 1.8}$ | $75.5_{\pm 0.9}$ | $96.8_{\pm 0.3}$ | $73.5_{\pm 0.6}$ | 82.6 |
| SAND-mask‡ (Shahtalebi et al., 2021) | $86.1_{\pm 0.6}$ | $80.3_{\pm 1.0}$ | $97.1_{\pm 0.3}$ | $80.0_{\pm 1.3}$ | 85.9 |
| DANN† (Ganin et al., 2016) | $86.4_{\pm 0.8}$ | $77.4_{\pm 0.8}$ | $97.3_{\pm 0.4}$ | $73.5_{\pm 2.3}$ | 83.7 |
| MTL† (Blanchard et al., 2021) | $87.5_{\pm 0.8}$ | $77.1_{\pm 0.5}$ | $96.4_{\pm 0.8}$ | $77.3_{\pm 1.8}$ | 84.6 |
| Mixup† (Xu et al., 2020) | $86.1_{\pm 0.5}$ | $78.9_{\pm 0.8}$ | $97.6_{\pm 0.1}$ | $75.8_{\pm 1.8}$ | 84.6 |
| MLDG† (Li et al., 2018a) | $85.5_{\pm 1.4}$ | $80.1_{\pm 1.7}$ | $97.4_{\pm 0.3}$ | $76.6_{\pm 1.1}$ | 84.9 |
| ERM† (Vapnik, 1999) | $84.7_{\pm 0.4}$ | $80.8_{\pm 0.6}$ | $97.2_{\pm 0.3}$ | $79.3_{\pm 1.0}$ | 85.5 |
| SagNet† (Nam et al., 2021) | $87.4_{\pm 1.0}$ | $80.7_{\pm 0.6}$ | $97.1_{\pm 0.1}$ | $80.0_{\pm 0.4}$ | 86.3 |
| Fishr† (Rame et al., 2022) | $87.9_{\pm 0.6}$ | $80.8_{\pm 0.5}$ | $97.9_{\pm 0.4}$ | $81.1_{\pm 0.8}$ | 86.9 |
| RDM (Nguyen et al., 2024) | $88.4_{\pm 0.2}$ | $81.3_{\pm 1.6}$ | $97.1_{\pm 0.1}$ | $81.8_{\pm 1.1}$ | 87.2 |
| Adam* (Kingma & Ba, 2015) | $88.0_{\pm 1.2}$ | $79.7_{\pm 0.5}$ | $96.7_{\pm 0.4}$ | $72.7_{\pm 0.9}$ | 84.3 |
| AdamW* (Loshchilov & Hutter, 2019) | $84.1_{\pm 1.5}$ | $80.7_{\pm 1.2}$ | $96.9_{\pm 0.4}$ | $72.8_{\pm 0.6}$ | 83.6 |
| SGD* (Nesterov, 1983) | $85.1_{\pm 0.4}$ | $76.0_{\pm 0.3}$ | $\mathbf{98.3}_{\pm 0.4}$ | $60.3_{\pm 6.1}$ | 79.9 |
| YOGI* (Zaheer et al., 2018) | $84.4_{\pm 1.7}$ | $79.7_{\pm 0.6}$ | $95.8_{\pm 0.3}$ | $65.1_{\pm 1.5}$ | 81.2 |
| AdaBelief* (Zhuang et al., 2020) | $85.4_{\pm 2.2}$ | $80.4_{\pm 1.1}$ | $97.4_{\pm 0.7}$ | $75.1_{\pm 1.4}$ | 84.6 |
| AdaHessian* (Yao et al., 2021) | $88.4_{\pm 0.6}$ | $80.0_{\pm 0.9}$ | $97.7_{\pm 0.4}$ | $71.7_{\pm 4.1}$ | 84.5 |
| SAM* (Foret et al., 2021) | $85.7_{\pm 1.2}$ | $81.0_{\pm 1.4}$ | $97.1_{\pm 0.2}$ | $77.4_{\pm 1.8}$ | 85.3 |
| GAM* (Zhang et al., 2023b) | $85.9_{\pm 0.9}$ | $81.3_{\pm 1.6}$ | $98.2_{\pm 0.4}$ | $79.0_{\pm 2.1}$ | 86.1 |
| SAGM* (Wang et al., 2023) | $87.6_{\pm 0.4}$ | $81.8_{\pm 0.5}$ | $97.2_{\pm 0.1}$ | $81.3_{\pm 2.1}$ | 86.9 |
| CRSAM* (Wu et al., 2024) | $86.7_{\pm 0.5}$ | $80.5_{\pm 1.6}$ | $96.4_{\pm 0.5}$ | $77.8_{\pm 2.4}$ | 85.4 |
| FSAM* (Li et al., 2024) | $87.6_{\pm 1.1}$ | $80.2_{\pm 0.7}$ | $95.4_{\pm 0.4}$ | $\mathbf{82.8}_{\pm 0.7}$ | 86.5 |
| **DA-SAM (ours)** | $\mathbf{88.6}_{\pm 0.8}$ | $\mathbf{82.9}_{\pm 0.4}$ | $96.7_{\pm 0.3}$ | $81.9_{\pm 1.2}$ | **87.5** |

*Table 14.* Full results ($mean_{\pm std}$) of our DA-SAM and existing DG methods calculated across three trials on VLCS. The results denoted by † are taken from (Wang et al., 2023) and (Dayal et al., 2023), while the results marked by ‡ are inherited directly from the original source. Results marked with * are inherited from (Chen et al., 2024b). The best results are highlighted in **bold**.

| Algorithm | Caltech | LabelMe | SUN | VOC | Average |
|---|---|---|---|---|---|
| MMD[†] (Li et al., 2018b) | $97.7_{\pm 0.1}$ | $64.0_{\pm 1.1}$ | $72.8_{\pm 0.2}$ | $75.3_{\pm 3.3}$ | 77.5 |
| Mixstyle[†] (Zhou et al., 2021) | $98.6_{\pm 0.3}$ | $64.5_{\pm 1.1}$ | $72.6_{\pm 0.5}$ | $75.7_{\pm 1.7}$ | 77.9 |
| GroupDRO[†] (Sagawa et al., 2020) | $97.3_{\pm 0.3}$ | $63.4_{\pm 0.9}$ | $69.5_{\pm 0.8}$ | $76.7_{\pm 0.7}$ | 76.7 |
| IRM[†] (Arjovsky et al., 2019) | $98.6_{\pm 0.1}$ | $64.9_{\pm 0.9}$ | $73.4_{\pm 0.6}$ | $77.3_{\pm 0.9}$ | 78.6 |
| ARM[†] (Zhang et al., 2021) | $98.7_{\pm 0.2}$ | $63.6_{\pm 0.7}$ | $71.3_{\pm 1.2}$ | $76.7_{\pm 0.6}$ | 77.6 |
| VREx[†] (Krueger et al., 2021) | $98.4_{\pm 0.3}$ | $64.4_{\pm 1.4}$ | $74.1_{\pm 0.4}$ | $76.2_{\pm 1.3}$ | 78.3 |
| AND-mask[‡] (Shahtalebi et al., 2021) | $98.3_{\pm 0.3}$ | $64.5_{\pm 0.2}$ | $69.3_{\pm 1.3}$ | $73.4_{\pm 1.3}$ | 76.4 |
| CDANN[†] (Li et al., 2018c) | $97.1_{\pm 0.3}$ | $65.1_{\pm 1.2}$ | $70.7_{\pm 0.8}$ | $77.1_{\pm 1.5}$ | 77.5 |
| SAND-mask[‡] (Shahtalebi et al., 2021) | $97.6_{\pm 0.3}$ | $64.5_{\pm 0.6}$ | $69.7_{\pm 0.6}$ | $73.0_{\pm 1.2}$ | 76.2 |
| MTL[†] (Blanchard et al., 2021) | $97.8_{\pm 0.4}$ | $64.3_{\pm 0.3}$ | $71.5_{\pm 0.7}$ | $75.3_{\pm 1.7}$ | 77.2 |
| Mixup[†] (Xu et al., 2020) | $98.3_{\pm 0.6}$ | $64.8_{\pm 1.0}$ | $72.1_{\pm 0.5}$ | $74.3_{\pm 0.8}$ | 77.4 |
| MLDG[†] (Li et al., 2018a) | $97.4_{\pm 0.2}$ | $65.2_{\pm 0.7}$ | $71.0_{\pm 1.4}$ | $75.3_{\pm 1.0}$ | 77.2 |
| ERM[†] (Vapnik, 1999) | $98.0_{\pm 0.3}$ | $64.7_{\pm 1.2}$ | $71.4_{\pm 1.2}$ | $75.2_{\pm 1.6}$ | 77.3 |
| SagNet[†] (Nam et al., 2021) | $97.9_{\pm 0.4}$ | $64.5_{\pm 0.5}$ | $71.4_{\pm 1.3}$ | $77.5_{\pm 0.5}$ | 77.8 |
| Fishr[†] (Rame et al., 2022) | $97.6_{\pm 0.7}$ | $67.3_{\pm 0.5}$ | $72.2_{\pm 0.9}$ | $75.7_{\pm 0.3}$ | 78.2 |
| RDM (Nguyen et al., 2024) | $98.1_{\pm 0.2}$ | $64.9_{\pm 0.7}$ | $72.6_{\pm 0.5}$ | $77.9_{\pm 1.2}$ | 78.4 |
| Adam* (Kingma & Ba, 2015) | $98.9_{\pm 0.4}$ | $65.9_{\pm 1.5}$ | $71.0_{\pm 1.6}$ | $74.5_{\pm 2.0}$ | 77.3 |
| AdamW* (Loshchilov & Hutter, 2019) | $98.3_{\pm 0.1}$ | $65.1_{\pm 1.7}$ | $70.9_{\pm 1.3}$ | $75.2_{\pm 1.5}$ | 77.4 |
| SGD* (Nesterov, 1983) | $98.4_{\pm 0.2}$ | $64.7_{\pm 0.7}$ | $72.5_{\pm 0.8}$ | $76.6_{\pm 0.8}$ | 78.1 |
| YOGI* (Zaheer et al., 2018) | $98.1_{\pm 0.7}$ | $63.9_{\pm 1.2}$ | $72.5_{\pm 1.6}$ | $75.7_{\pm 1.2}$ | 77.6 |
| AdaBelief* (Zhuang et al., 2020) | $98.0_{\pm 0.1}$ | $63.9_{\pm 0.4}$ | $73.4_{\pm 1.0}$ | $78.2_{\pm 1.8}$ | 78.4 |
| AdaHessian* (Yao et al., 2021) | $\mathbf{99.1}_{\pm 0.3}$ | $65.0_{\pm 1.7}$ | $72.7_{\pm 1.3}$ | $77.7_{\pm 1.0}$ | 78.6 |
| SAM* (Foret et al., 2021) | $98.5_{\pm 1.0}$ | $\mathbf{66.2}_{\pm 1.6}$ | $72.0_{\pm 1.0}$ | $76.1_{\pm 1.0}$ | 78.2 |
| GAM* (Zhang et al., 2023b) | $98.8_{\pm 0.6}$ | $65.1_{\pm 1.2}$ | $72.9_{\pm 1.0}$ | $77.2_{\pm 1.9}$ | 78.5 |
| SAGM* (Wang et al., 2023) | $98.4_{\pm 0.6}$ | $65.1_{\pm 1.1}$ | $74.1_{\pm 0.5}$ | $78.6_{\pm 2.4}$ | 79.1 |
| CRSAM* (Wu et al., 2024) | $99.0_{\pm 0.4}$ | $63.7_{\pm 0.8}$ | $73.8_{\pm 0.5}$ | $79.8_{\pm 1.4}$ | 79.1 |
| FSAM* (Li et al., 2024) | $98.9_{\pm 0.2}$ | $64.5_{\pm 0.4}$ | $74.1_{\pm 1.0}$ | $\mathbf{80.0}_{\pm 1.0}$ | 79.4 |
| **DA-SAM (ours)** | $98.7_{\pm 0.2}$ | $65.9_{\pm 1.3}$ | $\mathbf{75.2}_{\pm 0.8}$ | $78.8_{\pm 1.3}$ | **79.6** |

*Table 15.* Full results ($mean_{\pm std}$) of our DA-SAM and existing DG methods calculated across three trials on OfficeHome. The results denoted by † are taken from (Wang et al., 2023) and (Dayal et al., 2023), while the results marked by ‡ are inherited directly from the original source. Results marked with * are inherited from (Chen et al., 2024b). The best results are highlighted in **bold**.

| Algorithm | Art | Clipart | Product | Real-World | Average |
|---|---|---|---|---|---|
| MMD† (Li et al., 2018b) | $60.4_{\pm 0.2}$ | $53.3_{\pm 0.3}$ | $74.3_{\pm 0.1}$ | $77.4_{\pm 0.6}$ | 66.4 |
| Mixstyle† (Zhou et al., 2021) | $51.1_{\pm 0.3}$ | $53.2_{\pm 0.4}$ | $68.2_{\pm 0.7}$ | $69.2_{\pm 0.6}$ | 60.4 |
| GroupDRO† (Sagawa et al., 2020) | $60.4_{\pm 0.7}$ | $52.7_{\pm 1.0}$ | $75.0_{\pm 0.7}$ | $76.0_{\pm 0.7}$ | 66.0 |
| IRM† (Arjovsky et al., 2019) | $58.9_{\pm 2.3}$ | $52.2_{\pm 1.6}$ | $72.1_{\pm 2.9}$ | $74.0_{\pm 2.5}$ | 64.3 |
| ARM† (Zhang et al., 2021) | $58.9_{\pm 0.8}$ | $51.0_{\pm 0.5}$ | $74.1_{\pm 0.1}$ | $75.2_{\pm 0.3}$ | 64.8 |
| VREx† (Krueger et al., 2021) | $60.7_{\pm 0.9}$ | $53.0_{\pm 0.9}$ | $75.3_{\pm 0.1}$ | $76.6_{\pm 0.5}$ | 66.4 |
| AND-mask‡ (Shahtalebi et al., 2021) | $60.3_{\pm 0.5}$ | $52.3_{\pm 0.6}$ | $75.1_{\pm 0.2}$ | $76.6_{\pm 0.3}$ | 66.1 |
| CDANN† (Li et al., 2018c) | $61.5_{\pm 1.4}$ | $50.4_{\pm 2.4}$ | $74.4_{\pm 0.9}$ | $76.6_{\pm 0.8}$ | 65.7 |
| SAND-mask‡ (Shahtalebi et al., 2021) | $59.9_{\pm 0.7}$ | $53.6_{\pm 0.8}$ | $74.3_{\pm 0.4}$ | $75.8_{\pm 0.5}$ | 65.9 |
| DANN† (Ganin et al., 2016) | $59.9_{\pm 1.3}$ | $53.0_{\pm 0.3}$ | $73.6_{\pm 0.7}$ | $76.9_{\pm 0.5}$ | 65.9 |
| MTL† (Blanchard et al., 2021) | $61.5_{\pm 0.7}$ | $52.4_{\pm 0.6}$ | $74.9_{\pm 0.4}$ | $76.8_{\pm 0.4}$ | 66.4 |
| MLDG† (Li et al., 2018a) | $61.5_{\pm 0.9}$ | $53.2_{\pm 0.6}$ | $75.0_{\pm 1.2}$ | $77.5_{\pm 0.4}$ | 66.8 |
| ERM† (Vapnik, 1999) | $61.3_{\pm 0.7}$ | $52.4_{\pm 0.3}$ | $75.8_{\pm 0.1}$ | $76.6_{\pm 0.3}$ | 66.5 |
| SagNet† (Nam et al., 2021) | $63.4_{\pm 0.2}$ | $54.8_{\pm 0.4}$ | $75.8_{\pm 0.4}$ | $78.3_{\pm 0.3}$ | 68.1 |
| Fishr† (Rame et al., 2022) | $63.4_{\pm 0.8}$ | $54.2_{\pm 0.3}$ | $76.4_{\pm 0.3}$ | $78.5_{\pm 0.2}$ | 68.2 |
| RDM (Nguyen et al., 2024) | $61.1_{\pm 0.4}$ | $\mathbf{55.1}_{\pm 0.3}$ | $75.7_{\pm 0.5}$ | $77.3_{\pm 0.3}$ | 67.3 |
| Adam* (Kingma & Ba, 2015) | $63.9_{\pm 0.8}$ | $48.1_{\pm 0.6}$ | $77.0_{\pm 0.9}$ | $81.8_{\pm 1.6}$ | 67.6 |
| YOGI* (Zaheer et al., 2018) | $63.5_{\pm 1.0}$ | $49.2_{\pm 1.2}$ | $76.2_{\pm 0.5}$ | $\mathbf{84.5}_{\pm 0.6}$ | 68.3 |
| SAM* (Foret et al., 2021) | $63.5_{\pm 1.2}$ | $48.6_{\pm 0.9}$ | $77.0_{\pm 0.8}$ | $82.9_{\pm 1.3}$ | 68.0 |
| GAM* (Zhang et al., 2023b) | $63.0_{\pm 1.2}$ | $49.8_{\pm 0.5}$ | $\mathbf{77.6}_{\pm 0.6}$ | $82.4_{\pm 1.0}$ | 68.2 |
| **DA-SAM (ours)** | $\mathbf{64.2}_{\pm 0.7}$ | $53.8_{\pm 1.0}$ | $77.4_{\pm 0.2}$ | $78.1_{\pm 0.7}$ | **68.4** |

*Table 16.* Full results ($mean_{\pm std}$) of our DA-SAM and existing DG methods calculated across three trials on TerraIncognita. The results denoted by † are taken from (Wang et al., 2023) and (Dayal et al., 2023), while the results marked by ‡ are inherited directly from the original source. Results marked with * are inherited from (Chen et al., 2024b). The best results are highlighted in **bold**.

| Algorithm | L100 | L38 | L43 | L46 | Average |
|---|---|---|---|---|---|
| MMD† (Li et al., 2018b) | $41.9_{\pm 3.0}$ | $34.8_{\pm 1.0}$ | $57.0_{\pm 1.9}$ | $35.2_{\pm 1.8}$ | 42.2 |
| Mixstyle† (Zhou et al., 2021) | $54.3_{\pm 1.1}$ | $34.1_{\pm 1.1}$ | $55.9_{\pm 1.1}$ | $31.7_{\pm 2.1}$ | 44.0 |
| GroupDRO† (Sagawa et al., 2020) | $41.2_{\pm 0.7}$ | $38.6_{\pm 2.1}$ | $56.7_{\pm 0.9}$ | $36.4_{\pm 2.1}$ | 43.2 |
| IRM† (Arjovsky et al., 2019) | $54.6_{\pm 1.3}$ | $39.8_{\pm 1.9}$ | $56.2_{\pm 1.8}$ | $39.6_{\pm 0.8}$ | 47.6 |
| ARM† (Zhang et al., 2021) | $49.3_{\pm 0.7}$ | $38.3_{\pm 2.4}$ | $55.8_{\pm 0.8}$ | $38.7_{\pm 1.3}$ | 45.5 |
| VREx† (Krueger et al., 2021) | $48.2_{\pm 4.3}$ | $41.7_{\pm 1.3}$ | $56.8_{\pm 0.8}$ | $38.7_{\pm 3.1}$ | 46.4 |
| CDANN† (Li et al., 2018c) | $47.0_{\pm 1.9}$ | $41.3_{\pm 4.8}$ | $54.9_{\pm 1.7}$ | $39.8_{\pm 2.3}$ | 45.8 |
| DANN† (Ganin et al., 2016) | $51.1_{\pm 3.5}$ | $40.6_{\pm 0.6}$ | $57.4_{\pm 0.5}$ | $37.7_{\pm 1.8}$ | 46.7 |
| MTL† (Blanchard et al., 2021) | $49.3_{\pm 1.2}$ | $39.6_{\pm 6.3}$ | $55.6_{\pm 1.1}$ | $37.8_{\pm 0.8}$ | 45.6 |
| MLDG† (Li et al., 2018a) | $54.2_{\pm 3.0}$ | $\mathbf{44.3}_{\pm 1.1}$ | $55.6_{\pm 0.3}$ | $36.9_{\pm 2.2}$ | 47.8 |
| ERM† (Vapnik, 1999) | $54.3_{\pm 0.4}$ | $42.5_{\pm 0.7}$ | $55.6_{\pm 0.3}$ | $38.8_{\pm 2.5}$ | 47.8 |
| SagNet† (Nam et al., 2021) | $53.0_{\pm 2.9}$ | $43.0_{\pm 2.5}$ | $57.9_{\pm 0.6}$ | $40.4_{\pm 1.3}$ | 48.6 |
| RDM (Nguyen et al., 2024) | $52.9_{\pm 1.2}$ | $43.1_{\pm 1.0}$ | $58.1_{\pm 1.3}$ | $36.1_{\pm 2.9}$ | 47.5 |
| Adam* (Kingma & Ba, 2015) | $42.2_{\pm 3.4}$ | $40.7_{\pm 1.2}$ | $59.9_{\pm 0.2}$ | $35.0_{\pm 2.8}$ | 44.4 |
| AdamW* (Loshchilov & Hutter, 2019) | $44.2_{\pm 6.8}$ | $39.8_{\pm 1.9}$ | $60.3_{\pm 2.0}$ | $36.6_{\pm 1.8}$ | 45.2 |
| SGD* (Nesterov, 1983) | $41.8_{\pm 5.8}$ | $39.8_{\pm 3.9}$ | $\mathbf{60.5}_{\pm 2.2}$ | $37.5_{\pm 1.1}$ | 44.9 |
| YOGI* (Zaheer et al., 2018) | $43.9_{\pm 2.2}$ | $42.5_{\pm 2.6}$ | $60.5_{\pm 1.1}$ | $34.8_{\pm 1.6}$ | 45.4 |
| AdaBelief* (Zhuang et al., 2020) | $42.6_{\pm 6.7}$ | $43.0_{\pm 2.0}$ | $60.2_{\pm 1.3}$ | $35.1_{\pm 0.3}$ | 45.2 |
| AdaHessian* (Yao et al., 2021) | $42.5_{\pm 4.8}$ | $39.5_{\pm 1.0}$ | $58.4_{\pm 2.6}$ | $37.3_{\pm 0.8}$ | 44.4 |
| SAM* (Foret et al., 2021) | $42.9_{\pm 3.5}$ | $43.0_{\pm 2.2}$ | $60.5_{\pm 1.6}$ | $36.4_{\pm 1.2}$ | 45.7 |
| GAM* (Zhang et al., 2023b) | $42.2_{\pm 2.6}$ | $42.9_{\pm 1.7}$ | $60.2_{\pm 1.8}$ | $35.5_{\pm 0.7}$ | 45.2 |
| SAGM* (Wang et al., 2023) | $52.2_{\pm 3.9}$ | $42.3_{\pm 0.2}$ | $59.9_{\pm 0.3}$ | $39.9_{\pm 2.2}$ | 48.6 |
| CRSAM* (Wu et al., 2024) | $50.8_{\pm 0.5}$ | $35.3_{\pm 1.1}$ | $56.8_{\pm 0.1}$ | $38.4_{\pm 1.4}$ | 45.3 |
| FSAM* (Li et al., 2024) | $47.7_{\pm 3.1}$ | $39.3_{\pm 3.6}$ | $57.8_{\pm 1.7}$ | $39.5_{\pm 2.1}$ | 46.1 |
| **DA-SAM (ours)** | $\mathbf{59.0}_{\pm 0.6}$ | $39.2_{\pm 1.1}$ | $58.5_{\pm 1.9}$ | $\mathbf{43.3}_{\pm 1.5}$ | **50.0** |

*Table 17.* Full results ($mean_{\pm std}$) of our DA-SAM and existing DG methods calculated across three trials on DomainNet. The results denoted by † are taken from (Wang et al., 2023) and (Dayal et al., 2023), while the results marked by ‡ are inherited directly from the original source. Results marked with * are inherited from (Chen et al., 2024b). The best results are highlighted in **bold**.

| Algorithm | Clip | Info | Paint | Quick | Real | Sketch | Average |
|---|---|---|---|---|---|---|---|
| MMD† (Li et al., 2018b) | $32.1_{\pm13.3}$ | $11.0_{\pm4.6}$ | $26.8_{\pm11.3}$ | $8.7_{\pm2.1}$ | $32.7_{\pm13.8}$ | $28.9_{\pm11.9}$ | 23.4 |
| Mixstyle† (Zhou et al., 2021) | $51.9_{\pm0.4}$ | $13.3_{\pm0.2}$ | $37.0_{\pm0.5}$ | $12.3_{\pm0.1}$ | $46.1_{\pm0.3}$ | $43.4_{\pm0.4}$ | 34.0 |
| GroupDRO† (Sagawa et al., 2020) | $47.2_{\pm0.5}$ | $17.5_{\pm0.4}$ | $33.8_{\pm0.5}$ | $9.3_{\pm0.3}$ | $51.6_{\pm0.4}$ | $40.1_{\pm0.6}$ | 33.3 |
| IRM† (Arjovsky et al., 2019) | $48.5_{\pm2.8}$ | $15.0_{\pm1.5}$ | $38.3_{\pm4.3}$ | $10.9_{\pm0.5}$ | $48.2_{\pm5.2}$ | $42.3_{\pm3.1}$ | 33.9 |
| ARM† (Zhang et al., 2021) | $49.7_{\pm0.3}$ | $16.3_{\pm0.5}$ | $40.9_{\pm1.1}$ | $9.4_{\pm0.1}$ | $53.4_{\pm0.4}$ | $43.5_{\pm0.4}$ | 35.5 |
| VREx† (Krueger et al., 2021) | $47.3_{\pm3.5}$ | $16.0_{\pm1.5}$ | $35.8_{\pm4.6}$ | $10.9_{\pm0.3}$ | $49.6_{\pm4.9}$ | $42.0_{\pm3.0}$ | 33.6 |
| AND-mask‡ (Shahtalebi et al., 2021) | $52.3_{\pm0.8}$ | $17.3_{\pm0.5}$ | $43.7_{\pm1.1}$ | $12.3_{\pm0.4}$ | $55.8_{\pm0.4}$ | $46.1_{\pm0.8}$ | 37.9 |
| CDANN† (Li et al., 2018c) | $54.6_{\pm0.4}$ | $17.3_{\pm0.1}$ | $43.7_{\pm0.9}$ | $12.1_{\pm0.7}$ | $56.2_{\pm0.4}$ | $45.9_{\pm0.5}$ | 38.3 |
| SAND-mask‡ (Shahtalebi et al., 2021) | $43.8_{\pm1.3}$ | $15.2_{\pm0.2}$ | $38.2_{\pm0.6}$ | $9.0_{\pm0.2}$ | $47.1_{\pm1.1}$ | $39.9_{\pm0.6}$ | 32.2 |
| DANN† (Ganin et al., 2016) | $53.1_{\pm0.2}$ | $18.3_{\pm0.1}$ | $44.2_{\pm0.7}$ | $11.8_{\pm0.1}$ | $55.5_{\pm0.4}$ | $46.8_{\pm0.6}$ | 38.3 |
| MTL† (Blanchard et al., 2021) | $57.9_{\pm0.5}$ | $18.5_{\pm0.4}$ | $46.0_{\pm0.1}$ | $12.5_{\pm0.1}$ | $59.5_{\pm0.3}$ | $49.2_{\pm0.1}$ | 40.6 |
| Mixup† (Xu et al., 2020) | $55.7_{\pm0.3}$ | $18.5_{\pm0.5}$ | $44.3_{\pm0.5}$ | $12.5_{\pm0.4}$ | $55.8_{\pm0.3}$ | $48.2_{\pm0.5}$ | 39.2 |
| MLDG† (Li et al., 2018a) | $59.1_{\pm0.2}$ | $19.1_{\pm0.3}$ | $45.8_{\pm0.7}$ | $13.4_{\pm0.3}$ | $59.6_{\pm0.2}$ | $50.2_{\pm0.4}$ | 41.2 |
| ERM† (Vapnik, 1999) | $62.8_{\pm0.4}$ | $20.2_{\pm0.3}$ | $50.3_{\pm0.3}$ | $13.7_{\pm0.5}$ | $63.7_{\pm0.2}$ | $52.1_{\pm0.5}$ | 43.8 |
| SagNet† (Nam et al., 2021) | $57.7_{\pm0.3}$ | $19.0_{\pm0.2}$ | $45.3_{\pm0.3}$ | $12.7_{\pm0.5}$ | $58.1_{\pm0.5}$ | $48.8_{\pm0.2}$ | 40.3 |
| Fishr† (Rame et al., 2022) | $58.3_{\pm0.5}$ | $20.2_{\pm0.2}$ | $47.9_{\pm0.2}$ | $13.6_{\pm0.3}$ | $60.5_{\pm0.3}$ | $50.5_{\pm0.3}$ | 41.8 |
| RDM (Nguyen et al., 2024) | $62.1_{\pm0.2}$ | $20.7_{\pm0.1}$ | $49.2_{\pm0.4}$ | $14.1_{\pm0.4}$ | $63.0_{\pm1.3}$ | $51.4_{\pm0.1}$ | 43.4 |
| Adam* (Kingma & Ba, 2015) | $63.0_{\pm0.3}$ | $20.2_{\pm0.4}$ | $49.1_{\pm0.1}$ | $13.0_{\pm0.3}$ | $62.0_{\pm0.4}$ | $50.7_{\pm0.1}$ | 43.0 |
| AdamW* (Loshchilov & Hutter, 2019) | $63.0_{\pm0.6}$ | $20.6_{\pm0.2}$ | $49.6_{\pm0.0}$ | $13.0_{\pm0.2}$ | $63.6_{\pm0.2}$ | $50.4_{\pm0.1}$ | 43.4 |
| SGD* (Nesterov, 1983) | $61.3_{\pm0.2}$ | $20.4_{\pm0.2}$ | $49.4_{\pm0.2}$ | $12.6_{\pm0.1}$ | $65.7_{\pm0.0}$ | $49.6_{\pm0.2}$ | 43.2 |
| YOGI* (Zaheer et al., 2018) | $63.3_{\pm0.1}$ | $20.6_{\pm0.1}$ | $50.1_{\pm0.3}$ | $13.2_{\pm0.3}$ | $62.8_{\pm0.1}$ | $51.0_{\pm0.2}$ | 43.5 |
| AdaHessian* (Yao et al., 2021) | $63.3_{\pm0.2}$ | $21.4_{\pm0.1}$ | $\mathbf{50.8}_{\pm0.3}$ | $13.6_{\pm0.1}$ | $\mathbf{65.7}_{\pm0.1}$ | $51.4_{\pm0.2}$ | 44.4 |
| SAM* (Foret et al., 2021) | $63.3_{\pm0.1}$ | $20.3_{\pm0.3}$ | $50.0_{\pm0.3}$ | $13.6_{\pm0.2}$ | $63.6_{\pm0.3}$ | $49.6_{\pm0.4}$ | 43.4 |
| GAM* (Zhang et al., 2023b) | $63.0_{\pm0.5}$ | $20.2_{\pm0.2}$ | $50.3_{\pm0.1}$ | $13.2_{\pm0.3}$ | $64.5_{\pm0.2}$ | $51.6_{\pm0.5}$ | 43.8 |
| CRSAM* (Wu et al., 2024) | $62.9_{\pm0.3}$ | $20.7_{\pm0.2}$ | $50.8_{\pm0.2}$ | $\mathbf{15.2}_{\pm0.4}$ | $62.5_{\pm0.4}$ | $\mathbf{53.5}_{\pm0.2}$ | 44.3 |
| **DA-SAM (ours)** | $\mathbf{63.4}_{\pm0.2}$ | $\mathbf{21.9}_{\pm0.2}$ | $50.6_{\pm0.6}$ | $14.2_{\pm0.6}$ | $64.4_{\pm0.3}$ | $52.7_{\pm0.7}$ | **44.5** |

