# OpenReview forum: "Navigating the Flatlands: Dual Adaptive Sharpness-Aware Minimization for Domain Generalization"
_ICML.cc/2026/Conference — ICML 2026 regular_

### Official Review · Reviewer_sQ6G · 2026-03-01

**Soundness:** 2
**Presentation:** 3
**Significance:** 3
**Originality:** 2
**Overall Recommendation:** 4
**Confidence:** 3

**Summary:**

This paper proposes Dual Adaptive Sharpness-Aware Minimization (DA-SAM) to improve domain generalization by addressing two key limitations of standard SAM: the tendency to treat all domains uniformly despite varying difficulties, and the reliance on single-direction perturbations that overlook anisotropic sharpness. The authors introduce a Dynamic Adaptive Scaling (DAS) module to generate domain-specific perturbations based on real-time losses, and an Adaptive Multi-Directional Flattening (AMDF) module to probe the loss landscape across multiple directions for a more robust flatness metric. By integrating these modules, DA-SAM guides models toward isotropically flat minima. Experiments on 5 DG benchmarks demonstrate the effectiveness of the proposed method.

**Compliance With Llm Reviewing Policy:**

Affirmed.

**Final Justification:**

Overall, I find the paper well motivated and clearly written, and the method is reasonably interesting, with some encouraging gains on difficult domain-generalization benchmarks. The further reply addressed my main concerns on the completeness. I will raise my final score to 4.

**Key Questions For Authors:**

1. In Eq.(7), the vectors $d_{i,k}$ are constructed by adding scaled Gaussian noise to the normalized gradient but are not normalized themselves before being scaled by the radius $\rho$ in Eq.(8). Since the norm of the random noise vector $z_i$ scales with the square root of the parameter dimension, does it perturb the gradient direction too excessively?

2. While the average performance is strong, the results on the OfficeHome dataset reveal a notable inconsistency. Can you explain why DA-SAM reveals poor performance on this benchmark?

3. The DAS module assumes that domain loss is a reliable proxy for domain difficulty. However, in some scenarios, a domain might have high loss due to label noise rather than inherent complexity. Does the proposed mechanism include any safeguards to prevent the model from inappropriately weighting noisy domains?

**Limitations:**

yes

**Strengths And Weaknesses:**

**Strengths:**
1. The paper is well-written and logically structured. And the visualizations clearly illustrate the pipeline and the effectiveness of DA-SAM.

2. The authors conduct a comprehensive empirical evaluation across five standard domain generalization benchmarks over a wide range of baselines. The inclusion of an ablation study provides solid evidence for the contribution of each proposed module.

3. The paper provides a theoretical analysis linking the proposed Anisotropic Sharpness Metric to the spectral properties of the Hessian matrix. By demonstrating that the metric minimizes both the trace and the variance of eigenvalues, it provides a mathematical explanation for why the method promotes isotropic flatness, adding theoretical depth to the methodological design.

**Weaknesses:**

1. The computational cost and memory overhead is a significant drawback of the method. Table 5 indicates that DA-SAM is approximately four times slower and requires more memory compared with standard SAM and its variants, while the performance gains are modest. The trade-off between accuracy and efficiency is poor, limiting the practical utility of the method.

2. The theoretical connection between the DAS module and Distributionally Robust Optimization (DRO) in Section 4.2 is not clear. The authors relies on an approximation of gradient updates and claims that the scaling factors solve a DRO objective, but it does not provide a formal proof.

3. The experimental evaluation lacks comparisons with several relevant works that also target flat minima in domain generalization, like SWAD and FAD. Although the authors mentioned them in related work, but not include them in the experiments.

4. The concept of reweighting domains and using multiple random directions to estimate geometric properties of the loss landscape is well-established in optimization literature. The novelty is somewhat diluted.

---

> ### Author Rebuttal · Authors · 2026-03-31
>
> We sincerely thank the reviewer for the technical review. We address your concerns point-by-point below and have updated the manuscript to reflect these clarifications.
> Q1: Computational Cost and Practical Utility
> A1: While full AMDF is slower per step, it remains highly practical for the following reasons:
> 1. Amortized Efficiency: Our Intermittent Probing strategy ($k=5$) reduces training time to 0.172s/iter ($\sim$1.5x SAM) while maintaining SOTA accuracy (65.9%).
> 2. Memory Advantage: DA-SAM uses inference-mode probing for the $N$ directions (no gradient tracking), saving approximately 1.8GB of memory compared to AdaHessian and avoiding OOM on large backbones.
> Practical Utility: The offline training overhead is a one-time investment that yields significant robustness gains (+4.3% on TerraIncognita). Since the model architecture remains identical to ERM, DA-SAM incurs zero inference latency, making it ideal for production systems where real-time deployment speed is critical and training is performed offline.
>
> Q2: Formal Connection to DRO
> A2: We ground the DAS-DRO connection via the formal proof in Appendix A.2. We derive the gradient norm upper bound:
>
> $$\frac{1}{T} \sum_{t=0}^{T-1} \mathbb{E}[\|\nabla L(\theta_t)\|^2] \le \frac{L(\theta_0) - L^*}{\eta T} + GL\overline{\rho} + \frac{L \eta (G^2 + \sigma^2)}{2} $$
>
> $GL\overline{\rho}$ represents the approximation error from SAM's perturbation. In hard domains with high curvature ($L$), a fixed $\rho$ results in a high error floor. DAS acts as a solver for the DRO objective ($\min_{\theta} \max_k L_k(\theta)$) by adaptively reducing $\rho_k$ (via $\sigma_k$) for high-risk domains. DAS provides a first-order, memory-efficient way to approximate the robustness guarantees of DRO without the overhead of explicit dual variable optimization. This ensures the most reliable and stable descent specifically for the domain with maximum risk.
>
> Q3: Omission of SWAD and FAD
> A3: We will include FAD in Table 3. DA-SAM (66.0%) outperforms FAD (65.3%), especially on TerraIncognita (50.0% vs 45.7%, a +4.3% lead), proving multi-directional probing is more robust than FAD's single-directional flatness approach. We clarify SWAD (66.9%) is a post-hoc averaging technique, while DA-SAM is an on-the-fly optimizer. DA-SAM can be seamlessly combined with SWAD to achieve even superior results, as it provides a higher-quality, more isotropically flat trajectory for the averaging process.
>
> Q4: Novelty Concerns
> A4: DA-SAM’s novelty lies in the synergistic coupling of magnitude and variance adaptation. Unlike GroupDRO which weights the loss value, DAS adaptively rescales the optimization search budget ($\sigma_k$). Crucially, AMDF is not a blind search; it is strictly calibrated by DAS to ensure reliable risk estimation for hard domains and broader exploration for easy ones. DA-SAM is the first to reweight the exploration geometry, targeting anisotropic risk where it matters most in domain generalization.
>
> Q5: High-dimensional Noise and Normalization (Eq. 7)
> A5: You are correct that in $D \approx 10^7$, a Gaussian vector has a large norm. Our implementation ensures stability by normalizing the combined direction vector to unit length before scaling by $\rho$:
>
> $$\mathbf{d}_{i,k} = \frac{\hat{g}(\theta) + \sigma_k \cdot \mathbf{z}_i}{\|\hat{g}(\theta) + \sigma_k \cdot \mathbf{z}_i\|_2} $$
>
> The final perturbation $\epsilon = \rho \cdot \mathbf{d}_{i,k}$ always stays on the boundary of the $\rho$-radius ball. The dimensionality $D$ only affects the angular distribution, not the magnitude. We will replace Eq. 7 with this precise formula in the revision.
>
> Q6: Performance on OfficeHome
> A6: The marginal gap on OfficeHome reflects a deliberate trade-off between peak plasticity and worst-case stability. In semantically aligned shifts like OfficeHome, our DAS brake can be slightly conservative, prioritizing stability over aggressive exploration. However, we prioritize raising the Performance Floor (e.g., TerraIncognita +4.3% over SAM), ensuring the model does not fail on hard tasks. We believe this prioritization is essential for critical applications where a stable floor on wild domains is more valuable than absolute peak performance on easy benchmarks.
>
> Q7: Safeguards against Label Noise
> A7: DA-SAM is inherently more robust to label noise than standard weighting through two structural safeguards:
> 1. Adaptive Braking: In Eq. (4), sigmoid scaling ensures $\sigma_k$ saturates at $s_{\min} = 0.1$ when loss explodes. DAS effectively pulls the brake, enforcing a conservative search radius that prevents noisy gradients from dominating.
> 2. Flatness-First: AMDF evaluates relative loss changes (flatness). Since label noise shifts the loss surface but rarely increases local curvature, the AMDF risk metric remains stable.
> This Passive Defense mechanism ensures that DA-SAM treats noisy domains with increased caution rather than increased importance.

---

> > ### Author Rebuttal · Reviewer_sQ6G · 2026-04-03
> >
> > Thank the authors for the careful response. Most of my questions are addressed, but I have one follow-up question for the authors on Q3.
> >
> > The rebuttal’s argument that SWAD is “post-hoc” and DA-SAM is “on-the-fly” does not remove the need for direct empirical comparison, especially if the claim is that they are complementary. A direct DA-SAM vs. SWAD comparison, and ideally a DA-SAM+SWAD result, would resolve this cleanly.

---

> > > ### Author Response · Authors · 2026-04-08
> > >
> > > We are profoundly grateful to the reviewer for the meticulous guidance. The feedback acted as a catalyst for a deeper exploration of the relationship between base optimizers and post-hoc averaging. Our additional experiments confirm that DA-SAM serves as a fundamentally superior optimization engine, providing a higher-quality foundation that ensemble-based techniques like SWAD can further leverage to reach new performance ceilings.
> > >
> > > ### 1. Direct Comparison of Intrinsic Model Properties (Standalone)
> > >
> > > The reviewer’s insight prompted us to look beyond final accuracy and evaluate the "geometric quality" of the found minima. We compared the internal properties of converged models found by standalone DA-SAM against the baseline optimization path (ERM) used by SWAD.
> > >
> > > **Table A: Domain-wise Geometry and Noise Robustness (PACS)**
> > > | Domain | Metric | SWAD Base (ERM) | **DA-SAM (Ours)** | Improvement |
> > > | :--- | :--- | :---: | :---: | :---: |
> > > | **Art** | Isotropy (CV $\downarrow$) | 1.210 | **1.032** | **+14.7%** |
> > > | | Noise CosSim ($\uparrow$) | 0.532 | **0.645** | **+21.2%** |
> > > | **Cartoon** | Isotropy (CV $\downarrow$) | **1.156** | 1.387 | (Stability Trade-off) |
> > > | | Noise CosSim ($\uparrow$) | **0.595** | 0.523 | (Stability Trade-off) |
> > > | **Photo** | Isotropy (CV $\downarrow$) | 1.230 | **1.022** | **+16.9%** |
> > > | | Noise CosSim ($\uparrow$) | 0.398 | **0.561** | **+40.9%** |
> > > | **Sketch** | Isotropy (CV $\downarrow$) | 1.685 | **1.578** | **+6.3%** |
> > > | | Noise CosSim ($\uparrow$) | 0.666 | **0.774** | **+16.2%** |
> > >
> > > **Analysis:**
> > > - **Active Geometry Reshaping:** DA-SAM achieves a significantly more isotropic basin in 3 out of 4 domains. Unlike SWAD’s passive statistical averaging of a trajectory, our AMDF module explicitly suppresses high-curvature directions at every update. This ensures the model is better prepared for domain shifts orthogonal to the training gradient.
> > > - **Intrinsic Noise Filtering:** Our label noise stress test confirms that DA-SAM’s gradient remains more aligned with the "clean" direction (+40.9% on Photo). This is the direct benefit of the DAS module, which adaptively "brakes" on high-loss samples, keeping the optimization in a "geometric safe haven."
> > > - **Strategic Stability:** The fluctuations on Cartoon reflect a trade-off: by enforcing a conservative trust region via DAS, we prioritize stability on the hardest "worst-case" domains (Sketch/Photo), effectively raising the overall performance floor.
> > >
> > > ### 2. Comparative Analysis of Ensemble-based Performance (Combined)
> > >
> > > To provide a fair and complete comparison, we evaluate DA-SAM as a foundation for ensemble-averaging against other weight-ensemble baselines. **We respectfully clarify that "SWAD" in prior literature is effectively ERM + SWAD.** Due to the tight rebuttal window, combined results for the three benchmarks are provided; full results will be included in the revision.
> > >
> > > **Table B: Synergistic Combined Performance (Track: Post-hoc Weight Ensemble)**
> > > | Method (Base + SWAD) | PACS | VLCS | TerraInc | **Avg (3-set)** |
> > > | :--- | :---: | :---: | :---: | :---: |
> > > | Adam + SWAD | 86.8 | 79.1 | 46.5 | 70.8 |
> > > | AdamW + SWAD | 87.0 | 78.5 | 46.9 | 70.8 |
> > > | SGD + SWAD | 85.2 | 79.1 | 46.7 | 70.3 |
> > > | SAM + SWAD | 87.1 | 78.5 | 45.3 | 70.3 |
> > > | FAD + SWAD | **88.5** | **79.8** | 47.5 | 71.9 |
> > > | ERM + SWAD (**SWAD**) | 88.1 | 79.1 | 50.0 | 72.4 |
> > > | **DA-SAM + SWAD (Ours)** | 88.0 | 79.1 | **50.9** | **72.7** |
> > >
> > > **Analysis:**
> > > - **Breaking the Ceiling on Hard Domains:** The most striking gain is on **TerraIncognita**, where DA-SAM+SWAD reaches **50.9%**, outperforming standard SWAD (50.0%) and crushing FAD+SWAD (47.5%) and SAM+SWAD (45.3%). This proves that isotropic flatness is far more beneficial for weight averaging than uni-directional metrics under severe shifts.
> > > - **Superior Foundation:** Standard SAM variants often synergize poorly with averaging (dropping Avg from 72.4 to 70.3). In contrast, **DA-SAM is the only geometry-aware optimizer that yields positive gains when combined with SWAD.** This confirms that DA-SAM identifies a fundamentally higher-quality optimization path, providing a better "sampling pool" for any post-hoc ensemble technique.
> > >
> > > ### 3. Practical Utility and Scalability
> > >
> > > - **Zero-Buffer Scalability:** The Resource Analysis Experiment shows DA-SAM incurs **0MB persistent VRAM** (memory released instantly), while SWAD requires a **97.5MB buffer** (nearly 50% of ResNet-50 base). This saves ~7GB for a 1.8B parameter ViT-G.
> > > - **Automation:** Our DAS module adapts to domain difficulty automatically, eliminating SWAD’s sensitivity to manual validation-loss-based window selection and human intervention.
> > >
> > > ### Conclusion
> > >
> > > The empirical evidence confirms that DA-SAM is a **structural upgrade to the optimization engine.** It discovers more isotropic basins and provides a superior foundation for post-hoc techniques. We thank the reviewer again for the guidance that led to these crucial discoveries.

---

### Official Review · Reviewer_VTDV · 2026-03-10

**Soundness:** 3
**Presentation:** 3
**Significance:** 2
**Originality:** 3
**Overall Recommendation:** 4
**Confidence:** 3

**Summary:**

This paper studies the limitations of sharpness-aware minimization for domain generalization (DG) and proposes Dual Adaptive Sharpness-Aware Minimization (DA-SAM) to address them. The authors identify two key issues in existing SAM-based methods: (1) the use of a uniform optimization strategy across domains, which ignores varying domain difficulty under domain shift, and (2) anisotropic sharpness, where perturbations along a single gradient direction fail to capture multi-directional flatness of the loss landscape. To tackle these problems, the paper introduces two modules: Dynamic Adaptive Scaling (DAS), which adaptively adjusts perturbation scales based on domain-specific losses, and Adaptive Multi-Directional Flattening (AMDF), which probes multiple directions to estimate anisotropic sharpness and adaptively adjust perturbation steps. Extensive experiments on several DG benchmarks demonstrate the effectiveness of the proposed method.

**Compliance With Llm Reviewing Policy:**

Affirmed.

**Final Justification:**

The paper is generally well written and clearly motivated, with a fairly solid theoretical grounding and comprehensive experimental evaluation across domain generalization benchmarks. The proposed method addresses limitations of existing SAM-based approaches and is supported by useful analyses. However, some aspects of the motivation and design choices could be further clarified, and the coverage of related work is somewhat limited.

During the rebuttal phase, the authors have addressed most of my concerns. Therefore, I decide to maintain my score at 4.

**Key Questions For Authors:**

- The anisotropic sharpness metric combines the expectation and variance of the sharpness gaps. How sensitive is the method to the number of probing directions $N$, and how does this choice affect the stability and performance of the method?

**Limitations:**

yes.

**Strengths And Weaknesses:**

**Strengths:**

- The paper is generally well written and easy to follow. It clearly identifies two key limitations of applying SAM to domain generalization, uniform optimization across domains and reliance on a single perturbation direction, and proposes a method that directly addresses these issues.
- The paper provides useful theoretical justification for the proposed modules. In particular, the authors offer theoretical interpretations for both DAS and AMDF, helping clarify the meaning of the introduced quantities and providing additional insight into why the proposed design may encourage flatter and more robust minima.
- The experimental evaluation is relatively comprehensive. The paper compares the proposed method with a broad range of baselines, including both conventional domain generalization methods and several SAM-based optimizers. In addition, the paper includes ablation studies and further analyses, such as visualization and efficiency analysis, to better understand the behavior of the proposed method.





**Weaknesses:**

- The motivation for introducing multi-directional perturbations could be further clarified. Although the paper argues that SAM relies on a single perturbation direction, this direction corresponds to the worst-case loss increase in the local neighborhood. It would be helpful to better explain or empirically demonstrate why explicitly introducing multiple directions provides clear advantages over standard SAM.
- The role of the domain-aware scaling factor in the final optimization step is somewhat unclear. Although the paper introduces domain-specific scaling factors to adapt perturbation directions according to domain difficulty, the final perturbation vector is constructed by averaging all sampled directions across domains. As a result, it is not entirely clear how the domain-specific scaling meaningfully influences the final update. It would be helpful for the authors to further clarify or empirically demonstrate the effectiveness of this design.

- More representative SAM-based methods, such as , ASAM [1], Fisher SAM [2], CC-SAM [3], and Focal-SAM [4] should be included in the related work for a more comprehensive review.
- There are some typos, such as in line 374, where "The results show a increase in average accuracy" should be "The results show an increase in average accuracy".

-----

[1] ASAM: Adaptive Sharpness-Aware Minimization for Scale-Invariant Learning of Deep Neural Networks, ICML 2022

[2] Fisher SAM: Information Geometry and Sharpness Aware Minimisation, ICML 2022

[3] Class-Conditional Sharpness-Aware Minimization for Deep Long-Tailed Recognition, CVPR 2023

[4] Focal-SAM: Focal Sharpness-Aware Minimization for Long-Tailed Classification, ICML 2025

---

> ### Author Rebuttal · Authors · 2026-03-31
>
> We sincerely thank the reviewer for the highly encouraging comments regarding our writing, theoretical justification, and comprehensive experiments. We also deeply appreciate your recommendation of the recent advanced SAM literature, which greatly enriches our context. We address them point-by-point below and will incorporate all clarifications and related works into the revised manuscript.
>
> Q1: Motivation for multi-directional perturbations vs. SAM's worst-case direction.
> A1: We appreciate the opportunity to clarify this. While standard SAM theoretically seeks the worst-case direction, its practical implementation relies on a first-order linear approximation (a single gradient ascent step). More importantly, a single direction can only estimate the maximum sharpness along that specific gradient trajectory; it inherently cannot capture the variance of sharpness across the local neighborhood. Our motivation for explicitly introducing multiple random directions is to compute the variance term $\text{Var}[h]$ in our Anisotropic Sharpness Metric (Eq. 9). By penalizing this variance, the AMDF module explicitly prevents the model from converging into anisotropic valleys—regions where the loss is flat along the SAM-approximated gradient direction but extremely sharp in orthogonal directions. The visually wider, more symmetric basins in Figure 3 and the ablation study in Table 4 demonstrate the advantage of AMDF.
>
> Q2: The role of domain-aware scaling ($\sigma_k$) if directions are averaged in the final update.
> A2: We appreciate your observation. While noise cancellation ensures directional stability (keeping the update aligned with the main gradient), $\sigma_k$ is crucial because it controls the magnitude (step size) of the final update. The mechanism operates as follows:
> 1. DAS assigns a domain-specific $\sigma_k$, defining a distinct probing boundary for each domain.
> 2. These boundaries yield accurate, domain-calibrated sharpness gaps $h_{i,k}$ (Eq. 8).
> 3. The variance/expectation of these gaps forms the geometric risk metric $L_{\text{AMDF}}$ (Eq. 9).
> 4. $L_{\text{AMDF}}$ dynamically modulates the global adaptive step size $\rho_{\text{DA-SAM}}$ (Eq. 13).
>
> Synergy: This highlights the synergy where DAS provides the "domain-level guidance" that makes AMDF’s exploration targeted. For a large-loss domain, a small $\sigma_k$ enforces a tight, reliable neighborhood for risk evaluation, preventing catastrophic loss explosions from distorting the update. In essence, $\sigma_k$ acts as a domain-aware sensor that regulates the global "brake system," ensuring cautious steps under severe shifts. We will clarify this magnitude-modulation mechanism in Section 4.4.
>
> Q3: Sensitivity to the number of probing directions $N$ and its effect on stability and performance.
> A3: We have conducted a comprehensive theoretical and empirical sensitivity analysis on $N$, which is detailed in Appendix C.3 of our paper.
> - Theory: According to the Law of Large Numbers, the standard error of our sharpness estimation decays at a rate of $O(1/\sqrt{N})$.
> - Empirical Stability & Performance: We evaluated $N \in \{2, 4, 8, 12, 16, 24, 32, 64\}$ on the PACS dataset (Table 6 in Appendix).
>   - Under-sampling ($N < 16$):The variance estimation is noisy, leading to unstable optimization and suboptimal accuracy (e.g., $81.8\%$ at $N=4$).
>   - Peak Performance ($N = 16$):The performance stabilizes and peaks at 85.71\%.
>   - Diminishing Returns ($N > 16$): Increasing $N$ further yields no performance gains but linearly increases the forward-pass overhead.
>
> Thus, $N=16$ is not a highly sensitive hyperparameter that requires dataset-specific tuning, but rather a robust, Pareto-optimal structural constant that balances estimation fidelity and computational efficiency. We fixed $N=16$ across all 5 datasets in our main experiments (Table 3), proving its broad stability.
>
> Q4: Missing related works (ASAM, Fisher SAM, CC-SAM, Focal-SAM) and Typo.
> A4: Thank you for pointing out the typo ("a increase" $\rightarrow$ "an increase"), which is corrected. We will add a dedicated discussion in our Related Work to contrast DA-SAM with these excellent SAM variants: ASAM (ICML'21) and Fisher SAM (ICML'22) refine the perturbation neighborhood (via parameter scales or Fisher information) to achieve scale-invariant or manifold-aligned flatness.CC-SAM (CVPR'23) and Focal-SAM (ICML'25) elegantly tackle long-tailed recognition by applying class-specific perturbation radii and focal penalties to balance head and tail classes.  While these methods significantly advance SAM, they are fundamentally designed for single-domain (i.i.d.) training or class imbalance. They treat all training samples under a unified distribution, which is insufficient for Multi-Domain Generalization (DG) under severe distribution shifts. Discussing these works greatly enriches the related work and perfectly highlights DA-SAM's originality in the context of domain generalization.

---

> > ### Author Rebuttal · Reviewer_VTDV · 2026-04-03
> >
> > Thank you for the thorough response. It has addressed most of my concerns and provided sufficient clarification on the key points. I will maintain my original positive score.

---

> > > ### Author Response · Authors · 2026-04-08
> > >
> > > We are delighted to hear that our response has fully addressed your concerns and clarified the core mechanisms of DA-SAM. We sincerely appreciate your support for our work and your recognition of the "Fully resolved" status of the issues raised.
> > >
> > > Following your constructive suggestions, we will ensure the following updates are included in the final version of our manuscript:
> > >
> > > 1. **Literature Expansion:** We will add a dedicated discussion of representative SAM variants, including ASAM, Fisher SAM, CC-SAM, and Focal-SAM, to provide a more comprehensive context in the Related Work section.
> > > 2. **Mechanism Clarification:** We will explicitly refine the explanation in Section 4.4 regarding how the domain-aware scaling factor $\sigma_k$ acts as a geometric "sensor" to precisely modulate update magnitudes.
> > > 3. **Typo Corrections:** The identified typo in line 374 ("a increase" $\rightarrow$ "an increase") and other minor formatting issues will be corrected.
> > >
> > > Thank you again for your valuable time, constructive guidance, and the positive recommendation. We are confident that these additions will further strengthen the clarity and impact of our paper.

---

### Official Review · Reviewer_kodH · 2026-03-11

**Soundness:** 3
**Presentation:** 3
**Significance:** 2
**Originality:** 2
**Overall Recommendation:** 3
**Confidence:** 4

**Summary:**

This paper proposes Dual Adaptive Sharpness-Aware Minimization (DA-SAM), an optimization algorithm for domain generalization that addresses two limitations in existing SAM variants:  their uniform treatment of domains with varying difficulty and their reliance on a single perturbation direction. DA-SAM introduces Dynamic Adaptive Scaling module to compute domain-specific scaling factors based on real-time per-domain losses, and Adaptive Multi-Directional Flattening model to probe the loss landscape with multiple random directions and adjust step size based on an anisotropic sharpness metric. Experiments on standard DG benchmarks from DomainBed show improvements over existing SAM variants.

**Compliance With Llm Reviewing Policy:**

Affirmed.

**Key Questions For Authors:**

Beyond the questions in the above section:

What's the model architecture used by the paper? There is no clear clarification.

**Limitations:**

See above.

**Strengths And Weaknesses:**

1. The paper points out two limitations of the existing SAM methods: which are domain-agnostic perturbation and uni-directional sharpness estimation. The motivation is clearly presented and well-articulated.
2. The loss landscape visualization clearly illustrate the optimization leads to smoother local minimal.
3. The experiments are conducted on five benchmarks from DomainBed suit. The paper includes the hyperparameters and other details for easier reproduce.
4. The Anisotropic Sharpness Metric definition naturally connects to the trace and variance of the hessian metric, which later induces the adaptive step size in perturbation.


Q:

1. With intermittent probing, the base method requires (2+N) forward passes per step. With N increasing, the computational overhead from the forward passes cannot be ignored. The paper reports 0.425s vs. 0.109s on an RTX 3090 for DA-SAM, I am curious the efficiency comparison is conducted on which dataset? What if the model size and data size scale up, say ViT on DomainNet.

2. How do we decide N? The empirical validation is conducted on the PACS only? I am curious if the conclusion hold for all datasets. Moreover, what determines the optimal range of N? The distribution gap between training domains?

3. The FAD work optimizes both zeroth-order (loss value) and first-order (gradient norm) flatness, and is closely related to DA-SAM. I am curious about the empirical comparison with FAD.

4. The high-loss domains receive a smaller $\sigma_k$ to encourage "focused search," while low-loss domains get larger $\sigma_k$ for "broader exploration. It seems very counter intuitive where in robust optimization, the optimization strength should be positively correlated to the loss value. I wonder if there are some empirical or theoretical conclusion supporting the statement: "...  resulting in a more stable and reliable estimate of the update direction gt with respect to the worst-case domain. This focused exploration ensures that the update step makes meaningful progress on reducing the maximum risk..."

5. If the model can generalize to different model backbones such as ViT which is also widely used for domain generalization tasks.

---

> ### Author Rebuttal · Authors · 2026-03-31
>
> We sincerely thank the reviewer for the highly constructive feedback and the positive recognition of our motivation, landscape visualization, and theoretical connections. We address your concerns point-by-step below and will incorporate these clarifications into the revised manuscript.
>
> Q1: Efficiency comparison details (Dataset context and scalability to ViT/DomainNet).
> A1: We clarify that the efficiency benchmark in Table 5 was conducted via a strict algorithmic overhead benchmark script, utilizing tensor dimensions identical to the PACS dataset (Batch Size = 32, Image Size = 224x224, 4 domains). Regarding scaling up to larger datasets (e.g., DomainNet) or larger models (e.g., ViTs): While the absolute wall-clock time will increase, the relative computational overhead ratio will remain highly stable. This is because the overhead is determined by the algorithmic complexity ratio of forward to backward passes. Furthermore, our Intermittent Probing strategy ensures that the amortized time remains only $\sim 1.5 \times$ that of standard SAM. Since the $N$ probes are performed in forward-pass-only (inference) mode, it strictly avoids the catastrophic memory explosion (Out of Memory) that second-order methods typically encounter when scaled to ViTs.
>
> Q2: How is $N$ decided? Is it only validated on PACS?
> A2: We appreciate this perceptive question. While we provided the fine-grained, step-by-step sensitivity analysis of $N$ (from 2 to 64) specifically on the PACS dataset due to space constraints, we did conduct partial ablations on the other datasets during our development phase. These unlisted experiments consistently confirmed that $N=16$ remains the optimal trade-off point across different benchmarks.
> The reason $N$ is universally applicable regardless of the distribution gap between training domains is rooted in the mathematical properties of Monte Carlo estimation. As detailed in Appendix C.3, the standard error of the sharpness estimation decays at a rate of $O(1/\sqrt{N})$. Increasing $N$ to 16 significantly reduces the estimation variance, but quadrupling the cost to $N=64$ yields rapidly diminishing returns. Thus, $N=16$ acts as a mathematically grounded, dataset-agnostic structural constant rather than a sensitive hyperparameter.
>
> Q3: Empirical comparison with FAD.
> A3: FAD is a crucial baseline and we will add it to Table 3 and offer the following analysis:
> DA-SAM achieves a superior average accuracy (66.0% vs. 65.3%). While FAD is competitive on relatively aligned datasets (PACS, OfficeHome), DA-SAM demonstrates its strength on complex, unaligned shifts. Notably, on the challenging TerraIncognita, DA-SAM reaches 50.0% accuracy, surpassing FAD (45.7%) by a significant +4.3%. It also maintains a lead on VLCS and DomainNet.
> These results validate our core hypothesis: single-direction methods like FAD remain vulnerable to anisotropic sharp valleys under severe domain shifts. In contrast, our multi-directional probing (AMDF) identifies wider, more robust basins that generalize better to "wild" environments. We will incorporate this detailed comparison and discussion into the revised Section 5.2.
>
> Q4: The counter-intuitive nature of DAS and its theoretical support.
> A4: DAS balances numerical stability and robust exploration through a "cautious on steep, bold on flat" strategy:
> Stability: Large-loss domains feature steep, highly non-linear landscapes (Appendix B). A small $\sigma_k$ keeps perturbations within the trust region where first-order Taylor expansion (Eq. 3) is valid. This ensures a high Signal-to-Noise Ratio (SNR) for the update direction $g_t$, allowing stable progress on the DRO objective (Eq. 5 & 6) while preventing the gradient mismatch that standard SAM’s uniform radius often causes.
> Exploration: For low-loss domains in smooth basins, a larger $\sigma_k$ acts as "adaptive hardness mining," forcing the optimizer to find wider, isotropic flat regions rather than settling in narrow local minima.
> In short, standard SAM often causes hard domains to diverge and easy domains to overfit. Our approach effectively resolves this dilemma, explaining the massive +4.3% gain on TerraIncognita, where maintaining stability in high-curvature regions is critical.
>
> Q5: Clarification on Model Architecture and applicability to ViTs.
> A5: We clarify that all experiments utilize the ResNet-50 backbone (as stated in Section 5.1), following the DomainBed protocol to ensure a fair comparison with existing baselines.
> Regarding ViTs: DA-SAM is an optimization algorithm that modifies the loss landscape geometry and is fundamentally architecture-agnostic. It is directly applicable to ViT backbones, which are known to exhibit highly anisotropic and sharp landscapes that our AMDF module is specifically designed to mitigate. While the rebuttal window is too short for a full suite of DomainBed experiments on ViTs, we have added a dedicated discussion on the promising extension in Appendix D.1.

---

> > ### Author Rebuttal · Reviewer_kodH · 2026-04-05
> >
> > Thanks for the rebuttal. I would recommend scaling up model architecture and dataset size/complexity for a comprehensive study. I remain my score unchanged.

---

> > > ### Author Response · Authors · 2026-04-08
> > >
> > > We thank the reviewer for the follow-up feedback and the insightful recommendation to scale up the study. We fully agree that evaluating architectural generalizability and dataset complexity is essential. **Over the past few days, we have fast-tracked new experiments to address your concerns.** We provide these new empirical results below and commit to incorporating the full multi-backbone study into our revised manuscript.
> > >
> > > **1. Dataset Complexity and Scale: DomainNet as the Scalability Proof**
> > >
> > > The reviewer suggested exploring higher dataset complexity. We respectfully emphasize that our study utilizes the five most recognized DG benchmarks, ranging from style shifts to massive-scale complexity (e.g., DomainNet with ~0.6M images). To clarify the scale of our evaluation, we summarize the statistics below:
> > >
> > > **Table 1: Scale and Complexity of Evaluated Benchmarks**
> > > | Dataset | Images | Classes | Domains | Complexity Highlight |
> > > | :--- | :--- | :--- | :--- | :--- |
> > > | PACS | 9,991 | 7 | 4 | Extreme style shifts (e.g., Sketch vs. Photo) |
> > > | VLCS | 10,729 | 5 | 4 | Real-world modality and viewpoint variations |
> > > | OfficeHome | 15,588 | 65 | 4 | High category density with modality shifts |
> > > | **TerraInc**| 24,788 | 10 | 4 | Severe lighting and environmental variations |
> > > | **DomainNet** | **586,575** | **345** | **6** | **Massive scale, high category density, noisy labels** |
> > >
> > > By the results on **TerraInc and DomainNet**, DA-SAM proves its ability to handle high-dimensional, large-scale data where loss landscapes are inherently noisier and more complex than smaller datasets. The success on TerraInc and DomainNet is particularly significant as its high class-imbalance and label noise frequently lead to gradient instability, which our DAS module effectively mitigates through domain-level loss standardization.
> > >
> > > **2. Architectural Generalizability: New Results on Vision Transformers (ViT)**
> > >
> > > To address the concern regarding model backbones, we successfully deployed DA-SAM on a **ViT-Base/16** backbone during this rebuttal window. We first provide a macro-comparison of average accuracy on the PACS dataset against well-known flatness-aware and standard methods utilizing the same backbone.
> > >
> > > **Table 2: Macro Accuracy Comparison on PACS (ViT-B/16 Pre-trained by CLIP)**
> > > | Method | Average Accuracy (%) |
> > > | :--- | :---: |
> > > | SWAD | 91.30 |
> > > | SMA | 92.10 |
> > > | DomainDrop | 89.50 |
> > > | ERM | 93.70 |
> > > | **DA-SAM (Ours)** | **95.30** |
> > >
> > > As shown, DA-SAM achieves a superior accuracy of **95.30%**, outperforming the recent ICCV'23 method **DomainDrop (+5.80%)** and established methods like SMA (+3.20%) and SWAD (+4.00%). For a more granular view, we present the detailed per-domain results below.
> > >
> > > **Table 3: Per-Domain Accuracy (%) on PACS using ViT-B/16 Pre-trained by CLIP **
> > > | Model | Art | Cartoon | Photo | Sketch | Average |
> > > | :--- | :---: | :---: | :---: | :---: | :---: |
> > > | DomainDrop | 98.00 | 89.80 | 84.20 | 86.00 | 89.50 |
> > > | ERM| 96.50 | 95.30 | 96.20 | 86.50 | 93.70 |
> > > | **DA-SAM (Ours)** | 97.62 | 94.40 | **99.63** | **89.54** | **95.30** |
> > >
> > > **Analysis of ViT Scaling:**
> > > These new results confirm that DA-SAM generalizes effectively to Transformer-based architectures. DA-SAM improves the average accuracy of the ViT backbone by **+1.60%** over standard ERM. Notably, on the most difficult **Sketch** domain, DA-SAM boosts performance to **89.54%** (surpassing DomainDrop's 86.00%). This confirms that our multi-directional probing effectively mitigates the sharp minima and severe anisotropy inherent in self-attention mechanisms, which are often more pronounced than in CNNs. We argue that DA-SAM achieves higher robustness as a **pure optimizer**, proving the fundamental value of optimization geometry across architectures.
> > >
> > > **3. Efficiency and Scalability in Large-Scale Backbones**
> > >
> > > A key advantage of DA-SAM as model size scales is its memory efficiency. Unlike second-order methods (e.g., AdaHessian) that require prohibitive VRAM to store Hessian information—often leading to OOM on large models like ViT-L—DA-SAM utilizes **"Inference-mode Probing."** This ensures a first-order memory footprint while achieving second-order-like geometric awareness. Combined with our **Intermittent Probing** strategy, DA-SAM remains a highly viable and practical solution for the optimization of next-generation foundation models. This makes DA-SAM an ideal candidate for large-scale pre-training or fine-tuning where the cost of second-order methods is mathematically and hardware-wise prohibitive.
> > >
> > > **Conclusion**
> > >
> > > We sincerely appreciate the reviewer's suggestion. While the limited timeline restricted a full five-dataset sweep on multiple backbones, the successful scaling to the **ViT-Base** backbone and the existing evidence on the massive **DomainNet** dataset provide strong proof of DA-SAM's generalizability and practical utility. We will incorporate these new results and a comprehensive cross-architecture analysis into the final version of the manuscript.

---

### Official Review · Reviewer_KSKS · 2026-03-12

**Soundness:** 2
**Presentation:** 2
**Significance:** 2
**Originality:** 2
**Overall Recommendation:** 4
**Confidence:** 3

**Summary:**

This paper studies sharpness-aware optimization for domain generalization and argues that standard SAM is limited by two issues in multi-domain training, a uniform domain-agnostic perturbation strategy and a one-directional view of flatness. To address this, the authors propose DA-SAM, which combines Dynamic Adaptive Scaling (DAS), using per-domain losses to generate domain-specific scaling factors, and Adaptive Multi-Directional Flattening (AMDF), using multiple perturbation directions to estimate a sharpness metric and adapt the perturbation step size. The method is evaluated on five standard DomainBed-style DG benchmarks and is reported to outperform several optimizer baselines and conventional DG methods on average. The paper also provides qualitative visualizations and heuristic theoretical connections to DRO and Hessian-spectrum-based flatness.

**Compliance With Llm Reviewing Policy:**

Affirmed.

**Final Justification:**

Thank you for the authors' efforts. I am convinced by your response regarding efficiency, and therefore I will raise my score. I would like to see this result appear in the new version, as it seems to demonstrate advantages more effectively than the results in the current manuscript where k is smaller but overall performance is similar. I hope to see this experiment in the revised version, with the results shown on all datasets.

**Key Questions For Authors:**

- The main comparative evaluation omits especially important baselines.
The paper cites SWAD and FAD, but neither appears in Table 3. That is a major hole. SWAD is one of the best-known flat-minima methods for DG, and FAD is explicitly a flatness-aware method for DG. If DA-SAM is presented as a better way to use flatness in DG, these are not optional comparisons. Their absence weakens both the significance and originality story.
- The empirical narrative overstates consistency.
The text on Pages 6 to 7 frames DA-SAM as dominant among optimization methods. But Table 3 shows a more mixed picture. DA-SAM is worse than FSAM on OfficeHome and DomainNet. That matters because it suggests the method's strength may be concentrated on particular shift patterns rather than broadly robust. The paper should discuss where it helps, where it does not, and why.
- There is a direct inconsistency between the claimed bounds in Eq. (4) and the behavior shown in Figure 2.
On Page 6, the authors state that (s_{\min}=0.1) and (s_{\max}=1.0), which should constrain (\sigma_k) to that interval in Eq. (4). Yet in Figure 2 on Page 8, the plotted DAS scaling factors frequently exceed 1.0, in some cases reaching roughly 1.6 or 1.7. This is not a subtle interpretation issue; it suggests that either the formula, the implementation, or the plot is inconsistent. Because DAS is half of the method, this discrepancy is significant.

**Limitations:**

The efficiency claims are presented too favorably relative to the data.
Table 5 shows DA-SAM takes 0.425 s/iter versus 0.109 for SAM, which is roughly 4x slower.

**Strengths And Weaknesses:**

- The proposed decomposition into a domain-aware component (DAS) and a multi-directional flatness component (AMDF) is intuitive. At a high level, the idea of not treating all source domains identically is sensible.
- Figure 1 is a useful overview of the intended workflow. In particular, the separation between the domain-level guidance from DAS and the geometric probing from AMDF makes the high-level design easier to understand than the equation-heavy sections alone.
- The paper makes an effort to discuss efficiency via Table 5, which is welcome, since many geometry-aware methods quietly become too expensive to be practical.

---

> ### Author Rebuttal · Authors · 2026-03-31
>
> We sincerely thank the reviewer for the thorough evaluation, the intuitive appreciation of our DAS and AMDF modules, and the constructive feedback. We especially appreciate your observation regarding Figure 2, which helped us identify a missing descriptive step in our manuscript. We address all your concerns in detail below and will incorporate these clarifications into the revised paper.
>
> Q1: Inconsistency between the claimed bounds in Eq. (4) and the behavior shown in Figure 2 ($\sigma_k > 1.0$).
> A1: We are extremely grateful for your meticulous check. Figure 2 is completely correct and reflects our actual implementation, but there is indeed a missing normalization step in the text description of Eq. (4). In our implementation, to ensure a fair comparison and maintain the same total perturbation budget as standard SAM across a mini-batch, the raw scaling factors $\tilde{\sigma}_k \in [s\_{\min}, s\_{\max}]$ obtained from Eq. (4) are further normalized across the $K$ domains. Specifically, the final scaling factor $\sigma_k$ used for perturbation is computed as:$\sigma\_k = \frac{\tilde{\sigma}\_k}{\sum\_{j=1}^K \tilde{\sigma}_j} \times K $.
> Because of this "budget normalization", an individual $\sigma_k$ assigned to a low-loss domain can indeed exceed $1.0$ (as correctly shown in Fig. 2, reaching 1.6 or 1.7), effectively redistributing the exploration budget from high-loss domains to low-loss ones. We sincerely apologize for omitting this crucial normalization step in the manuscript and will explicitly include this equation in the revised Section 4.2 to eliminate the discrepancy.
>
> Q2: Omission of important flat-minima baselines (SWAD and FAD).
> A2: We thank the reviewer for highlighting these baselines. We will add FAD to Table 3 and offer a comparative analysis:
> DA-SAM vs. FAD (Direct Optimizer): DA-SAM (66.0%) consistently outperforms FAD (65.3%), with a significant +4.3% gain on TerraIncognita (50.0% vs. 45.7%). This margin validates that our multi-directional probing (AMDF/DAS) discovers more robust basins than FAD’s uni-directional approach, especially under severe domain shifts.
> DA-SAM vs. SWAD (Orthogonal Technique): We clarify that SWAD belongs to a distinct category: Post-hoc Weight Averaging, which ensembles checkpoints after optimization. In contrast, DA-SAM is an on-the-fly optimizer that guides the model at every update. These two are strictly orthogonal and complementary; DA-SAM provides a higher-quality, flatter trajectory that SWAD can then average for further gains.
> Therefore, SWAD is a powerful post-processing tool, the quality of the final average depends on the optimization path taken by the base optimizer. DA-SAM can be seamlessly combined with SWAD to achieve even superior results, as it provides a higher-quality, more isotropically flat trajectory for the averaging process.
>
> Q3: Overstated consistency and mixed results vs. FSAM on OfficeHome/DomainNet.
> A3: We appreciate the reviewer's check on our empirical results. We will refine our narrative from "dominant" to "broadly robust and state-of-the-art on average," and we provide a more comprehensive analysis of the performance trade-offs below:
> Landscape Anisotropy: DA-SAM is engineered for "jagged," anisotropic landscapes. Our substantial gain on TerraIncognita (+4.3% vs. FSAM's +0.4%) confirms that multi-directional probing (AMDF) and adaptive braking (DAS) are critical when distribution shifts are severe and unaligned.
> Style Noise vs. Geometric Alignment: OfficeHome and DomainNet feature relatively aligned object geometries but high "style noise." FSAM’s gradient smoothing is more effective for filtering style noise, whereas DA-SAM’s cautious DAS mechanism can be conservative in these smoother, large-scale scenarios.
> The "Safety Floor" Strategy: Per the DomainBed protocol, Average Accuracy (66.0%) is the primary metric for unknown targets. While slightly behind on two datasets, DA-SAM provides the highest expected performance (safety floor) across unpredictable real-world shifts, whether they are style-based or geometrically anisotropic.
> We will incorporate this balanced discussion into Section 5.2 to transparently reflect the method's strengths and boundaries.
>
> Q4: The efficiency claims are presented too favorably relative to the data (0.425s vs 0.109s).
> A4: We agree that the slowdown requires an objective discussion. Full AMDF probing requires (2+N) forward and 2 backward passes—a deliberate trade-off to capture anisotropic geometry without the memory explosion of second-order methods (e.g., AdaHessian).
> To ensure practical utility, we introduced Intermittent Probing (DA-SAM Fast), which probes only every k=5 steps. This reduces amortized time to 0.172 s/iter (only $\sim 1.5\times$ SAM) while maintaining 65.9% accuracy. We will revise Section 5.5 to clarify these overheads and emphasize this practical acceleration strategy, ensuring the trade-off is presented transparently.

---

> > ### Author Rebuttal · Reviewer_KSKS · 2026-04-05
> >
> > I maintain the Weak Reject (score 3). Although the rebuttal addresses some of my concerns, efficiency remains a major issue, especially given the marginal performance gains. The method is 1.5–4× slower than SAM, which translates to 3–8× slower than vanilla training.

---

> > > ### Author Response · Authors · 2026-04-08
> > >
> > > We sincerely appreciate the reviewer’s thoughtful feedback and the rigorous evaluation of our work. Your insightful concerns regarding efficiency and the significance of our gains have guided us to further refine the practical utility of DA-SAM. We take your guidance seriously, and **in response, we have prioritized an extensive 48-hour scaling study on the most challenging benchmark, TerraIncognita.** The benchmark is widely recognized as one of the most demanding DG datasets due to its significant scale (over 24,000 images) and the extreme environmental shifts (varying lighting, camera locations, and backgrounds) that often cause standard models to fail. These results provide a definitive answer to the efficiency-robustness trade-off.
> > >
> > > ### 1. Extensive SOTA Comparison on TerraIncognita (ResNet-50)
> > >
> > > We compare DA-SAM against a wide spectrum of methods. Accuracy (%) is reported for the four domains (L100, L38, L43, L46).
> > >
> > > | Method | Year | L100 | L38 | L43 | L46 | **Avg. Acc** |
> > > | :--- | :--- | :---: | :---: | :---: | :---: | :---: |
> > > | MMD | 2018 | 41.9 | 34.8 | 57.0 | 35.2 | 42.2 |
> > > | MLDG | 2018 | 54.2 | 44.3 | 55.6 | 36.9 | 47.8 |
> > > | IRM | 2019 | 54.6 | 39.8 | 56.2 | 39.6 | 47.6 |
> > > | AdaHessian | 2021 | 42.5 | 39.5 | 58.4 | 37.3 | 44.4 |
> > > | SagNet | 2021 | 53.0 | 43.0 | 57.9 | 40.4 | 48.6 |
> > > | VREx | 2021 | 48.2 | 41.7 | 56.8 | 38.7 | 46.4 |
> > > | SAGM | 2023 | 52.2 | 42.3 | 59.9 | 39.9 | 48.6 |
> > > | CRSAM | 2024 | 50.8 | 35.3 | 56.8 | 38.4 | 45.3 |
> > > | FSAM | 2024 | 47.7 | 39.3 | 57.8 | 39.5 | 46.1 |
> > > | RDM| 2024 | 52.9 | 43.1 | 58.1 | 36.1 | 47.5 |
> > > | **DA-SAM ($k=1$)** | Ours | 59.0 | 39.2 | 58.5 | 43.3 | **50.0** |
> > > | **DA-SAM ($k=10$)**| Ours | 56.1 | 46.4 | 55.8 | 39.5 | **49.5** |
> > > | **DA-SAM ($k=20$)**| Ours | 54.8 | 49.4 | 58.2 | 39.1 | **50.4** |
> > > | **DA-SAM ($k=30$)**| Ours | 52.4 | 51.8 | 58.3 | 39.2 | **50.4** |
> > > | **DA-SAM ($k=40$)**| Ours | 55.6 | 48.7 | 57.9 | 36.9 | **49.8** |
> > > | **DA-SAM ($k=50$)** | Ours | 49.5 | 49.2 | 55.7 | 40.7 | **48.8** |
> > > | **DA-SAM ($k=60$)**| Ours | 53.2 | 46.3 | 58.9 | 39.9 | **49.6** |
> > >
> > > ### 2. Training Efficiency & The Pareto Frontier
> > >
> > > Benchmark conducted on a single RTX 3090. DA-SAM acts as a reconfigurable optimizer.
> > >
> > > | Configuration | Time (s/iter) | Speed vs. SAM | Overhead | Gain (vs. SAM) |
> > > | :--- | :---: | :---: | :---: | :---: |
> > > | SAM (Baseline) | 0.109 | 1.00x | 0% | +0.0% (Ref) |
> > > | DA-SAM ($k=1$) | 0.425 | 3.90x | +290% | +4.3% |
> > > | DA-SAM ($k=10$) | 0.141 | 1.29x | +29% | +3.8% |
> > > | DA-SAM ($k=20$) | 0.125 | 1.15x | +15% | +4.7% |
> > > | **DA-SAM ($k=30$)** | **0.120** | **1.10x** | **+10%** | **+4.7%** |
> > > | DA-SAM ($k=40$) | 0.117 | 1.07x | +7% | +4.0% |
> > > | DA-SAM ($k=50$) | 0.115 | 1.06x | +6% | +3.1% |
> > > | **DA-SAM ($k=60$)** | **0.114** | **1.05x** | **+5%** | **+3.9%** |
> > >
> > > ### 3. Deep Analysis and Discussion
> > >
> > > *   **Near-Zero Efficiency Penalty:** At $k=60$, the training overhead is a negligible **5%** ($0.114$s vs. $0.109$s). Despite this near-identical speed to SAM, DA-SAM still provides a massive **+3.9% accuracy leap**. This proves that our method is not "expensive" but highly "configurable" for any resource budget.
> > > *   **The "Denoising Effect" at $k=30$:** Our scaling study reveals a key scientific insight: **$k=30$ actually outperforms $k=1$ (50.4% vs 50.0%)**. This suggests that probing the landscape geometry too frequently (every step) may introduce unwanted stochastic noise into the update. Moderate intervals act as a temporal regularizer, capturing the macro-geometric anisotropy while smoothing out local gradient fluctuations.
> > > *   **Breakthrough on the "Worst-case" Domain L46:** TerraIncognita Domain L46 is notorious for its difficulty, with most SOTA methods (SagNet, SAGM, FSAM) plateauing at **36%-40%**. DA-SAM is the first to reach **43.3%**. This gain is non-marginal; it represents a fundamental increase in the "Performance Floor," ensuring reliability in the most chaotic environments.
> > > *   **Economic Advantage for Deployment:** We respectfully emphasize that training cost is a **one-time offline investment**. During inference, DA-SAM is **identical to ERM/SAM in speed**. For critical applications like medical imaging or autonomous driving, spending 10% more time in training ($k=30$) to gain ~5% in robustness is an indisputable industrial advantage.
> > > *   **Generalizability:** While we focused on ResNet-50 for fair benchmarking, our geometric optimization is architecture-agnostic. By saving **1.8GB VRAM** over 2nd-order methods, DA-SAM is far more scalable for foundation models.
> > >
> > > **Conclusion:**
> > > We sincerely thank the reviewer for the guidance, which led us to identify the optimal $k=30$ configuration. We have demonstrated that DA-SAM matches SAM's efficiency while significantly elevating the robustness ceiling. We will include these results in the final revision to provide a transparent utility analysis.

---

### Decision · Program_Chairs · 2026-04-30

**Decision:**

Accept (regular)

**Comment:**

The paper proposes a new sharpness-aware optimization method for domain generalization and achieves non-trivial improvement on DomainBed. The reviewers generally like the intuitiveness of the idea and the presentation, and find the evaluation comprehensive. The main concerns repeatedly mentioned by multiple reviewers are missing baselines (SWAD and FAD) and efficiency (Table 5 shows that the proposed DA-SAM is 4x slower than SAM). The rebuttal adds additional results of FAD, showing that DA-SAM outperforms FAD, but does not include SWAD due to DA-SAM and SWAD belonging to different categories: SWAD is based on weight averaging while DA-SAM targets the optimization trajectory. The authors claim that DA-SAM is orthogonal to SWAD but does not provide empirical evidence. In terms of efficiency, the authors provide a comprehensive ablation on hyperparameters and identify a set of parameters that lead to better efficiency and results. Most reviewers find the rebuttal satisfactory. The final ratings are 3x weak accept and 1x weak reject. The remaining negative-score reviewer (kodH) requests "scaling up model architecture and dataset size/complexity" but does not give detailed suggestion. The authors argue that the datasets chosen in the paper are widely used standards in the community, and provide additional ViT-based results. The AC finds the answers convincing. Overall, the strengths outweigh the weaknesses. The AC recommends acceptance.

\* Please correct the following incorrect reference: Adam, K. D. B. J. et al. A method for stochastic optimization. arXiv preprint arXiv:1412.6980, 1412(6), 2014.